# Revisiting Logistic-softmax Likelihood in Bayesian Meta-learning for Few-shot Classification

**Tianjun Ke**[†][*]**, Haoqun Cao**[†][*]**, Zenan Ling**[‡]**, Feng Zhou**[†][§]
[†]Center for Applied Statistics and School of Statistics, Renmin University of China
[‡]School of EIC, Huazhong University of Science and Technology
{keanson, caohaoqun2007, feng.zhou}@ruc.edu.cn, lingzenan@hust.edu.cn

## Abstract

Meta-learning has demonstrated promising results in few-shot classification (FSC) by learning to solve new problems using prior knowledge. Bayesian methods are effective at characterizing uncertainty in FSC, which is crucial in high-risk fields. In this context, the logistic-softmax likelihood is often employed as an alternative to the softmax likelihood in multi-class Gaussian process classification due to its conditional conjugacy property. However, the theoretical property of logistic-softmax is not clear and previous research indicated that the inherent uncertainty of logistic-softmax leads to suboptimal performance. To mitigate these issues, we revisit and redesign the logistic-softmax likelihood, which enables control of the *a priori* confidence level through a temperature parameter. Furthermore, we theoretically and empirically show that softmax can be viewed as a special case of logistic-softmax and logistic-softmax induces a larger family of data distribution than softmax. Utilizing modified logistic-softmax, we integrate the data augmentation technique into the deep kernel based Gaussian process meta-learning framework, and derive an analytical mean-field approximation for task-specific updates. Our approach yields well-calibrated uncertainty estimates and achieves comparable or superior results on standard benchmark datasets. Code is publicly available at https://github.com/keanson/revisit-logistic-softmax.

## 1 Introduction

Meta-learning refers to the ability to quickly learn a new task given a set of training tasks that share a common structure [13; 15; 30]. This concept is critical for achieving human-like performance computationally with little data, and recent algorithms have shown success in few-shot classification and regression problems [15; 18]. When dealing with limited data, it is essential to analyze robust meta-learning methods that properly deal with uncertainty which is inevitable in the context [1]. Some existing works have suggested that the Bayesian inference mechanism provides a principled way to address this issue, moving towards robust meta-learning with uncertainty quantification [7; 27; 38].

The Bayesian framework combines a prior distribution and a likelihood function that models the data distribution. By performing posterior inference on parameters, it provides a natural framework for capturing inherent model uncertainty [8; 23]. Gaussian process (GP) is a Bayesian nonparametric method that places a prior distribution on functions rather than parameters, ensuring better expressiveness [3]. The GP prior is defined by a covariance function which can be parameterized by deep neural networks, commonly referred to as a deep kernel [35]. GP with a deep kernel has demonstrated state-of-the-art performance in few-shot regression problems [20], where posterior inference admits a closed-form solution due to the conjugacy between the Gaussian likelihood and the GP prior.

---

[*]Equal contributions.
[§]Corresponding author.

37th Conference on Neural Information Processing Systems (NeurIPS 2023).

However, there are several challenges in GP-based meta-learning with deep kernels for classification tasks. To begin with, unlike the regression scenario, the widely-used softmax likelihood does not lead to conjugacy for GPs, making posterior inference intractable. To overcome this issue, several approximate inference methods have been proposed, such as label regression with Gaussian likelihood [20], One-vs-Each (OVE) approximation of softmax [27], and the logistic-softmax likelihood [9]. The logistic-softmax likelihood replaces the exponential function in the softmax likelihood with a logistic function, ensuring conditional conjugacy after data augmentation [9]. However, previous research suggests that the logistic-softmax function tends to exhibit inherent lack of confidence, leading to suboptimal performance in few-shot classification tasks [27]. Additionally, most GP-based meta-learning models employ Gibbs sampling for posterior inference, which can be computationally demanding for achieving convergence [26]. Furthermore, the coordination of task-level posterior inference and meta-level optimization for deep kernel methods remains an open question.

In this paper, we bridge these gaps by redesigning the logistic-softmax likelihood and deriving a mean-field approximation for posterior inference. In contrastive learning, adding a temperature parameter to rescale the logits in softmax is a prevalent approach [4; 12; 33]. Motivated by this concept, we introduce the temperature parameter to the logistic-softmax likelihood to control its inherent prior confidence. Moreover, we discover that softmax can be viewed as a special case of logistic-softmax and logistic-softmax induces a larger family of data distribution than softmax. To the best of our knowledge, this theoretical property has not been covered in the literature. Furthermore, we apply the modified logistic-softmax likelihood to GP based meta-learning for few-shot classification. Since the logistic-softmax likelihood enables a conditional conjugate GP model, we naturally derive an analytical mean-field approximation for task-specific updates. Compared to the existing literature [27], this variational inference method is more efficient than Gibbs sampling. We empirically show that our mean-field approximation achieves comparable results to the Gibbs sampling in practice. We also contribute to the coordination problem of bi-level optimization in Bayesian meta-learning methods.

Specifically, our contributions are as follows: (1) we introduce the logistic-softmax with temperature and prove its unique limiting behavior, which may have broad applicability across diverse machine learning domains; (2) we theoretically and empirically show that softmax can be viewed as a particular case of logistic-softmax and logistic-softmax induces a larger family of data distribution than softmax; (3) we derive an analytical mean-field approximation for task-level posterior inference with redesigned logistic-softmax through data augmentation; (4) we demonstrate the effectiveness of the redesigned logistic-softmax through few-shot classification accuracy and uncertainty calibration on several benchmark datasets; (5) we contribute to the coordination of task-level inference and meta-level optimization that appears in Bayesian meta-learning, which may provide insights for future research.

## 2 Preliminaries

**Meta-learning** involves learning from a set of tasks in order to acquire knowledge and generalize to new tasks [5; 20]. In this work, the $t$-th task comprises a dataset $D_t$ consisting of both support and query data, denoted as $\mathcal{S}_t$ and $\mathcal{Q}_t$ respectively. The support data $\mathcal{S}_t$ consists of a limited number of samples ($N$-shot), represented as pairs of input-output $(x_n, y_n)$, while the query set $\mathcal{Q}_t$ typically includes a larger number of samples. During training, models are exposed to various tasks sampled from the dataset $D$ to learn and extract valuable information across different contexts. Subsequently, when presented with an unseen task $t^*$, the model aims to leverage the acquired knowledge from both training data and the support set $\mathcal{S}_{t^*}$ to make accurate predictions on the query set $\mathcal{Q}_{t^*}$.

**Gaussian Process Classification** is a probabilistic framework used for solving classification problems. At its core, a GP is a probability distribution over functions, where the values of the function $f(x)$ evaluated at an arbitrary set of inputs have a joint Gaussian distribution with a mean vector and a covariance matrix determined by a kernel function. In the context of a $C$-class classification problem, separate GP latent functions $\{f^c\}_{c=1}^C$ are employed to model the logits for each class. These latent functions represent the underlying mapping from the input space to the logit of each class, capturing the uncertainty associated with the classification decision. By modeling the class logits with GPs, it enables principled probabilistic inference and prediction.

**Deep Kernel** is proposed by Wilson et al. [35] which combines kernel methods and neural networks, harnessing the expressive power from both sides. This deep kernel extends traditional covariance kernels by integrating a deep architecture into the base kernel formulation. The deep kernel is defined

as $k(\mathbf{x}, \mathbf{x}' \mid \boldsymbol{\theta}, \mathbf{w}) = k'(g_{\mathbf{w}}(\mathbf{x}), g_{\mathbf{w}}(\mathbf{x}') \mid \boldsymbol{\theta})$, where $k'$ represents the base kernel with parameters $\boldsymbol{\theta}$. To enhance flexibility, the inputs $\mathbf{x}$ and $\mathbf{x}'$ are transformed by a deep neural network $g$ with weights $\mathbf{w}$. The deep kernel parameters consist of $\boldsymbol{\theta}$ and $\mathbf{w}$. A notable advantage of deep kernels is their capability to learn metrics through data-driven optimization of input space transformation, in contrast to traditional kernels that often rely on Euclidean distance-based metrics.

# 3 Logistic-softmax with Temperature

In this section, we introduce a novel logistic-softmax function that incorporates a temperature parameter. We first emphasize two unique features of this function as the temperature approaches zero. We then demonstrate that logistic-softmax surpasses softmax as a versatile categorical likelihood theoretically and empirically. Specifically, we show that softmax can be viewed as a special case of logistic-softmax and logistic-softmax induces a larger family of data distributions than softmax. Fig. 1 briefly previews the relationship between the two likelihoods.

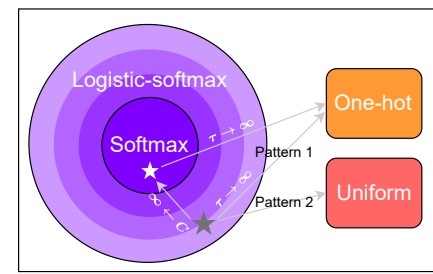

## 3.1 Definition of Logistic-softmax with Temperature

We define the logistic-softmax function with temperature as:

Figure 1: Diagram representing the features and relationship of logistic-softmax and softmax.

$$p(y = k \mid \mathbf{f}_n) = \frac{\sigma(f_n^k/\tau)}{\sum_{c=1}^{C} \sigma(f_n^c/\tau)}, \tag{3.1}$$

where we assume $C$ classes, $f_n^c = f^c(\mathbf{x}_n)$, $\mathbf{f}_n = [f_n^1, \ldots, f_n^C]^\top$, $k \in [C] := \{1, \ldots, C\}$, $\tau$ is the temperature parameter and $\sigma(\cdot)$ is the logistic function as opposed to the exponential in the traditional softmax. While there are other possible choices for incorporating the idea of temperature, we find this particular form is the most straightforward one while preserving the desired conditional conjugate property of logistic-softmax as shown in later sections.

Although reminiscent of the softmax likelihood with temperature, the logistic-softmax likelihood displays distinct limiting behavior, illustrated by the following theorem.

**Theorem 3.1.** Denote the logistic-softmax function with temperature as $\mathrm{LS}(\mathbf{f}_n, \tau)$. Define $I := \{i : f_n^i > 0\} \subset [C]$, we have

$$\lim_{\tau \to 0^+} \mathrm{LS}(\mathbf{f}_n, \tau) = \begin{cases} \boldsymbol{e}_{c^*}, & \text{if } \max_{c \in [C]} f_n^c < 0 \text{ and } c^* = \operatorname*{argmax}_{c \in [C]} f_n^c \\ \dfrac{1}{|I|} \sum_{c \in I} \boldsymbol{e}_c, & \text{if } \max_{c \in [C]} f_n^c > 0 \end{cases}$$

where $\boldsymbol{e}_c \in \mathbb{R}^C$ is the one-hot vector with a 1 in its $c$-th coordinate.

The proof is provided in Appendix I. On the one hand, the logistic-softmax likelihood converges to a one-hot vector when $f_n^c < 0$ for all $c \in [C]$. On the other hand, the likelihood transforms into a uniform distribution over the classes corresponding to the indices of the positive elements if multiple elements of $\mathbf{f}_n$ are greater than 0.

We interpret the theoretical result from two perspectives. First, the original logistic-softmax likelihood has been criticized for its lack of confidence [27]. However, with the introduction of temperature, the modified logistic-softmax likelihood can become excessively confident as the temperature approaches zero. This implies that we can effectively control the confidence of the logistic-softmax likelihood by adjusting the temperature, thus resolving existing issues. Second, when multiple components of the logits are positive, this likelihood exhibits remarkable fairness to all positive classes. This characteristic is profound and distinct since it enables adaptation to multi-label classification scenarios, where assigning an equal probability to each correct class is essential. Notably, to our knowledge, no existing likelihood can identify multiple correct classes, which limits the tools available for multi-label classification. Within this domain, the prevalent approach is to treat each class as a binary classification problem separately. Our discovery paves the way for a new paradigm in this domain, as

the logistic-softmax function allows for the identification of several positive labels simultaneously. However, in this work, we primarily concentrate on the application of logistic-softmax in Bayesian meta-learning and postpone the investigation of multi-label classification to future research.

## 3.2 Comparison of Logistic-softmax and Softmax with Temperature

We present several results demonstrating that logistic-softmax surpasses softmax as a versatile categorical likelihood function theoretically and empirically. We begin by giving a theorem that shows the logistic-softmax function converges to the softmax function pointwise.

**Theorem 3.2.** For all $\mathbf{f}_n \in \mathbb{R}^C$, $\tau \in \mathbb{R} \setminus \{0\}$ and $C_0 \in \mathbb{R}$, we have

$$\lim_{C' \to +\infty} \mathrm{LS}(\mathbf{f}_n - C', \tau) = \mathrm{S}(\mathbf{f}_n, \tau) = \mathrm{S}(\mathbf{f}_n - C_0, \tau),$$

where $\mathrm{S}(\mathbf{f}_n, \tau)$ denotes the softmax function with temperature.

The proof is provided in Appendix II. This theorem shows that softmax is a translational invariant function that can be approximated by logistic-softmax with a sufficient negative shift of logits. We also note that logistic-softmax is not translational invariant. Empirically, we find that $C' = 5$ provides an accurate approximation if $\mathbf{f}_n$ is near zero. Now we present a stronger result indicating that logistic-softmax is more versatile than softmax with regards to modeling the data distribution.

**Theorem 3.3.** Assume the logits $f^c \sim \mathcal{GP}(a, k^c)$, where $a$ is the mean function and $k^c$ is the kernel function for each class $c \in [C]$. Denote $\mathbf{y} = [y_1, \ldots, y_N]^\top$ as the random label vector of $N$ given points. Suppose $a \in \mathscr{A}$ and $k^c \in \mathscr{K}$, where $\mathscr{A}$ and $\mathscr{K}$ are two function classes. Define $\mathscr{F}(\ell \mid \mathscr{A}, \mathscr{K})$ as the family of the marginal distribution $p(\mathbf{y}|\mathbf{X}, a, k^c)$ induced by $a \in \mathscr{A}$ and $k^c \in \mathscr{K}$ on given points $\mathbf{X} \in \mathbb{R}^{N \times p}$ with a likelihood function $\ell$. Under mild condition on $\mathscr{A}$, we have

$$\mathscr{F}(\mathrm{S} \mid \mathscr{A}, \mathscr{K}) = \mathscr{F}(\mathrm{S} \mid \mathscr{K}).$$

Furthermore, we have

$$\mathscr{F}(\mathrm{S} \mid \mathscr{A}, \mathscr{K}) \subset \mathscr{F}(\mathrm{LS} \mid \mathscr{K}).$$

The proof is provided in Appendix III. We elaborate on this theoretical result below. The first equation indicates that data distribution induced by softmax is translational invariant with regards to $a$, while the one by logistic-softmax is not. Recall that Theorem 3.2 implies softmax is translational invariance w.r.t. its input variable. Intuitively, as the location parameter $a$ is integrated due to translational invariance, the family of the marginal distribution of $y$ corresponding to softmax is only decided by the kernel function $k$, while the family for logistic-softmax has an additional parameter $a$. With this intuition, it should be rather straightforward to understand our second result. Informally, the second result states that logistic-softmax has a larger family than softmax. In other words, logistic-softmax possesses a stronger capability in modeling the data distribution for each class.

Next, we delve deeper into the relationship between logistic-softmax and softmax. For comparison, we present a toy classification task with three classes ($C = 3$) and one sample belonging to the first class ($y = 1$). We place a standard normal prior on $f_1$ and $f_2$ and clamp $f_3$ at $-100$ for ease of visualization. We plot the likelihoods in Fig. 2 where we observe a distinct pattern between softmax and logistic-softmax. Although both likelihoods become more confident (likelihood output changes at a faster rate) as the temperature parameter decreases, logistic-softmax exhibits unique probability patterns when both $f_1$ and $f_2$ are greater than 0, while softmax gives the same sets of borderline parallel to $f_1 - f_2 = 0$ regardless of location. We present the zoom-in plots to further explain the difference in Fig. 2. These plots demonstrate that when both $f_1$ and $f_2$ are a bit less than 0, logistic-softmax accurately approximates softmax in every temperature. Indeed, the pattern of softmax in the whole plane is the same as the pattern that appeared in quadrant 3 of logistic-softmax. This observation supports the idea behind Theorem 3.2 and Theorem 3.3, which shows that logistic-softmax is capable of modeling any data distribution that can be modeled by softmax, as long as we adjust the logits to the negative domain or put a sufficiently small negative mean on $f$. Furthermore, as Fig. 2 shows, the softmax likelihood is unable to model the data distribution of logistic-softmax with positive logits. Therefore, we conclude that logistic-softmax is more powerful than softmax for its expressiveness in modeling categorical distribution.

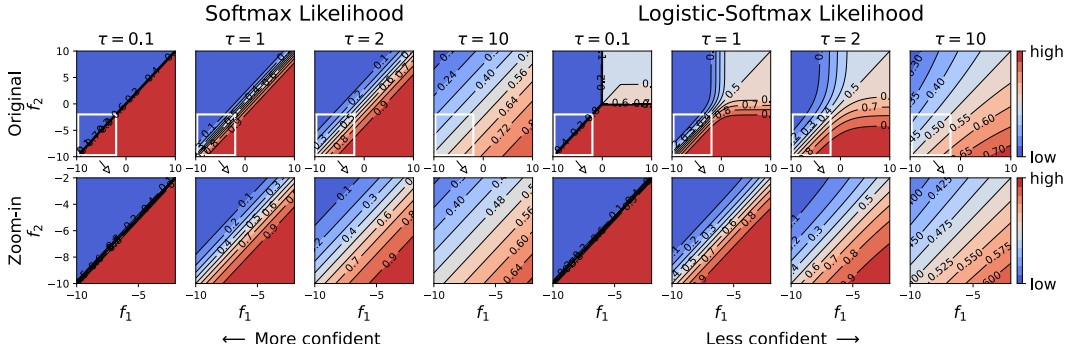

Figure 2: Plot of $p(y = 1|\mathbf{f})$ where $f_3$ clamped to $-100$. We provide separate zoom-in plots of softmax and logistic-softmax in the 2nd row. In the upper-right area (where all $f_1$ and $f_2$ are greater than 0), the logistic-softmax function exhibits unique probability patterns that softmax cannot model. In the bottom-left area (where all $f_1$ and $f_2$ are smaller than 0), logistic-softmax accurately approximates softmax for every temperature and every location all at once.

## 3.3 Further Discussions of Potential Applications

As the logistic-softmax function possesses more expressiveness than the softmax function, we include a short discussion of its potential applications in various domains. To begin with, it can be applied to Gaussian process classification tasks, including class-imbalanced scenarios [37], active learning [40], and time-series data analysis [6]. In this case, logistic-softmax brings desired conditional conjugacy to make inference tractable and provides additional flexibility in data modeling than softmax as indicated in our paper. Moreover, the logistic-softmax function can be a great choice in modern Bayesian methods, such as Bayesian neural networks and neural network Gaussian processes although further adaptation is needed. Furthermore, logistic-softmax might be capable of replacing softmax beyond the Bayesian domain since we prove its flexibility over the softmax function. For example, as the logistic-softmax function captures positive signals for multiple classes, it may have prospective advantages in scenarios like multi-label classification [14] and multi-label contrastive learning [39], ushering in new paradigms thanks to its unique properties.

## 4 Logistic-softmax with Temperature in Bayesian Meta-learning

Logistic-softmax was initially proposed to address the conjugation issue that arises in multi-class Gaussian process classification. As the Bayesian framework is advantageous in uncertainty calibration, some researchers focus on adapting the multi-class Gaussian process to few-shot classification tasks, where Bayesian meta-learning is one of the prevalent paradigms. In specific, Snell & Zemel [27] attempts to apply original logistic-softmax to Bayesian meta-learning but fails to provide efficient inference and optimal result. To follow this research line and validate the advantages of logistic-softmax with temperature, we utilize the logistic-softmax likelihood in the GP-based meta-learning framework. Additionally, we use the data augmentation technique to derive a fully analytical mean-field inference method for this model.

### 4.1 Framework of Bayesian Meta-learning

We review the Bayesian meta-learning framework used in the research line of MAML [7], DKT [20], and OVE [27]. To begin with, Bayesian meta-learning commonly utilizes a hierarchical structure that includes an inner loop and an outer loop. Conventionally, task-specific parameters are updated in the inner loop and task-common parameters are updated in the outer loop [7]. For efficiency, DKT proposed to replace the inner loop with a marginal likelihood computation that integrates out task-specific GP function for each task [22]. Then, the task-common hyperparameter $\boldsymbol{\Theta}$ of the deep kernel can be learned by maximizing the marginal likelihood across all tasks in the outer loop. More specifically, denote the input support and query data of task $t$ as $D_t^x$, the target data as $D_t^y$. $D^x$ and

$D^y$ are the collections of these datasets over all tasks. The marginal likelihood takes the form

$$p(D^y \mid D^x, \boldsymbol{\Theta}) = \prod_t \int p(D_t^y \mid D_t^x, \boldsymbol{\phi}_t) p(\boldsymbol{\phi}_t \mid \boldsymbol{\Theta}) d\boldsymbol{\phi}_t, \tag{4.1}$$

where $\boldsymbol{\phi}_t$ is the task-specific parameters of task $t$. To obtain an analytic integral of Eq. (4.1), DKT used a GP prior for $p(\boldsymbol{\phi}_t \mid \boldsymbol{\Theta})$. The integral can be computed analytically in regression cases since the model likelihood is Gaussian as opposed to classification cases where the conjugacy is broken since the likelihood is Bernoulli or categorical. However, we can use the logistic-softmax likelihood to obtain a GP model that is conditional conjugate after data augmentation. Thus, an efficient inference method is needed for the augmented model. We describe our task-level inference in next section.

## 4.2 Task-level Bayesian Inference

In this subsection, we introduce how an efficient task-level Bayesian inference method is developed. For a specific task, we denote the (support and query) input dataset as $\mathbf{X} = [\mathbf{x}_1, \dots, \mathbf{x}_N]^\top$ with label dataset $\mathbf{y} = [y_1, \dots, y_N]^\top$ where $y_n \in [C]$, $N$ and $C$ are the number of observations and classes respectively. The multi-class GP classification model includes latent GP functions for each class $\mathbf{f} = [f^1, \dots, f^C]^\top$ where $f^c \sim \mathcal{GP}(a^c, k^c)$, $a^c$ is mean function and $k^c$ is kernel for $c$-th class.

We utilize the logistic-softmax function with temperature in Eq. (3.1) to model the likelihood for the multi-class GP classification. Inspired by Galy-Fajou et al. [9], three sets of auxiliary latent variables are augmented to expand the logistic-softmax likelihood to obtain a conditional conjugate model for each task, including Gamma variables $\boldsymbol{\lambda}$, Poisson variables $\mathbf{M}$, and Pólya-Gamma variables $\boldsymbol{\Omega}$. Given the GP priors on $f^c$, we obtain the augmented joint density (proof provided in Appendix IV):

$$p(\mathbf{Y}, \boldsymbol{\lambda}, \mathbf{M}, \boldsymbol{\Omega}, \mathbf{F}) = \prod_{n=1}^N \prod_{c=1}^C 2^{-(y_n^c + m_n^c)} \exp\left( \frac{y_n^c - m_n^c}{2} \frac{f_n^c}{\tau} - \frac{\omega_n^c}{2} \left(\frac{f_n^c}{\tau}\right)^2 \right) \mathrm{PG}(\omega_n^c \mid m_n^c + y_n^c, 0)$$
$$\cdot \frac{\lambda_n^{m_n^c}}{m_n^c!} \exp(-\lambda_n) \cdot \prod_{c=1}^C \mathcal{N}(\mathbf{f}^c \mid \mathbf{a}^c, \mathbf{K}^c), \tag{4.2}$$

where $\mathbf{f}^c = [f_1^c, \dots, f_N^c]^\top$ is the $c$-th column of $\mathbf{F}$, $\lambda_n$ is the augmented Gamma variable, $m_n^c$ is the augmented Poisson variable, $\omega_n^c$ is the augmented Pólya-Gamma variable, $\mathbf{a}^c$ is the mean and $\mathbf{K}^c$ is the kernel matrix for $c$-th class with $N$ samples. The original model has been now transformed into a conditional conjugate one, which possesses excellent mathematical properties.

**Mean-field Approximation** Based on Eq. (4.2), we can obtain the closed-form conditional densities for all variables, which constitutes a Gibbs sampler if we iteratively draw a sample from each conditional distribution. However, the Gibbs sampler is not efficient because the sampling operation is time-consuming. In order to improve efficiency, we derive a mean-field variational inference that provides an approximate posterior but with better efficiency.

In the mean-field algorithm, we need to approximate the true posterior $p(\boldsymbol{\lambda}, \mathbf{M}, \boldsymbol{\Omega}, \mathbf{F} \mid \mathbf{Y})$ by a variational distribution which is assumed to factorize over some partition of latent variables. Here, we assume the variational distribution $q(\boldsymbol{\lambda}, \mathbf{M}, \boldsymbol{\Omega}, \mathbf{F}) = q_1(\mathbf{M}, \boldsymbol{\Omega}) q_2(\boldsymbol{\lambda}, \mathbf{F})$. Following the standard derivation provided in Appendix V, we obtain the optimal density for each factor:

$$q_1(\boldsymbol{\Omega}|\mathbf{M}) = \prod_{n,c=1}^{N,C} \mathrm{PG}(\omega_n^c \mid m_n^c + y_n^c, \widetilde{f}_n^c), \quad \text{(4.3a)} \qquad q_2(\boldsymbol{\lambda}) = \prod_{n=1}^N \mathrm{Ga}(\lambda_n \mid \alpha_n, C), \qquad \text{(4.3c)}$$

$$q_1(\mathbf{M}) = \prod_{n,c=1}^{N,C} \mathrm{Po}(m_n^c \mid \gamma_n^c), \qquad \text{(4.3b)} \qquad q_2(\mathbf{F}) = \prod_{c=1}^C \mathcal{N}(\mathbf{f}^c \mid \widetilde{\boldsymbol{\mu}}^c, \widetilde{\boldsymbol{\Sigma}}^c), \qquad \text{(4.3d)}$$

where

$$\widetilde{f}_n^c = \frac{1}{\tau}\sqrt{\mathbb{E}[f_n^{c2}]} = \frac{1}{\tau}\sqrt{\widetilde{\mu}_n^{c2} + \widetilde{\sigma}_{nn}^{c2}}, \quad \text{(4.4a)}$$

$$\widetilde{\mathbf{\Sigma}}^c = (\mathrm{diag}(\bar{\omega}_n^c/\tau^2) + \mathbf{K}^{c^{-1}})^{-1}, \quad \text{(4.4d)}$$

$$\gamma_n^c = \frac{\exp(\psi(\alpha_n) - \widetilde{\mu}_n^c/2\tau)}{2C\cosh(\widetilde{f}_n^c/2)}, \quad \text{(4.4b)}$$

$$\widetilde{\boldsymbol{\mu}}^c = \frac{1}{2\tau}\widetilde{\mathbf{\Sigma}}^c(\mathbf{y}^c - \boldsymbol{\gamma}^c) + \widetilde{\mathbf{\Sigma}}^c\mathbf{K}^{c^{-1}}\mathbf{a}^c, \quad \text{(4.4e)}$$

$$\alpha_n = \sum_{c=1}^{C}\gamma_n^c + 1, \quad \text{(4.4c)}$$

$$\bar{\omega}_n^c = \mathbb{E}[\omega_n^c] = \frac{\gamma_n^c + y_n^c}{2\widetilde{f}_n^c}\tanh\frac{\widetilde{f}_n^c}{2}. \quad \text{(4.4f)}$$

where $\psi(\cdot)$ is the digamma function. Update the posterior of $\mathbf{\Omega}, \mathbf{M}, \boldsymbol{\lambda}$ and $\mathbf{F}$ iteratively by Eq. (4.3), we obtain an efficient mean-field algorithm to provide the approximate posterior for each task.

## 4.3 Meta-level Optimization

We learn a set of hyperparameters $\mathbf{\Theta}$ of deep kernels in the outer loop that maximizes the marginal likelihood across all tasks, which is also called the empirical Bayes [17]:

$$\sum_{t=1}^{T}\log p(\mathbf{Y}_t \mid \mathbf{\Theta}) = \sum_{t=1}^{T}\log \int p(\mathbf{Y}_t, \boldsymbol{\lambda}_t, \mathbf{M}_t, \mathbf{\Omega}_t, \mathbf{F}_t \mid \mathbf{\Theta})d\boldsymbol{\lambda}_t d\mathbf{M}_t d\mathbf{\Omega}_t d\mathbf{F}_t, \quad \text{(4.5)}$$

where the subscript $t$ indicates the $t$-th task. Unfortunately, the integral in Eq. (4.5) is intractable, so we utilize an approximate approach: given the variational parameters of each task, we try to maximize the evidence lower bound (ELBO) $\mathcal{L}$ as a function of $\mathbf{\Theta}$ [16]. Thanks to the data augmentation, we have an analytical expression of the ELBO which is provided in Appendix VI, and the gradient $\nabla_{\mathbf{\Theta}}\mathcal{L}$ can be computed by the automatic differentiation [2]. In addition, Snell & Zemel [27] has found that the predictive likelihood (PL) can also serve as a loss of $\mathbf{\Theta}$. We follow their derivation and use the approximate gradient estimator:

$$\nabla_{\boldsymbol{\theta}}\mathcal{L}_{\mathrm{PL}} \approx \frac{1}{M}\sum_{m=1}^{M}\nabla_{\boldsymbol{\theta}}\log p(y_* = k \mid \mathbf{x}_*, \mathbf{X}, \mathbf{Y}, \widehat{\mathbf{\Theta}}),$$

where $M$ denotes the number of samples drawn from Eq. (4.6b).

Alternating between the optimization steps w.r.t. the variational parameters of each task in the inner loop and the hyperparameters of deep kernels in the outer loop, we obtain an efficient Bayesian meta-learning method for classification. However, the coordination of the gradient flow between the inner loop and the outer loop remains an open question. Previous research detaches the task-level variables in the inner loop and updates the hyperparameters of the deep kernel using gradients produced by the outer loop [27]. Nevertheless, we have discovered that the task-level variables in the inner loop should not be detached when using mean-field approximation.

## 4.4 Prediction

Given a test task with support dataset containing the input $\mathbf{X} = \{\mathbf{x}_l\}_{l=1}^{L}$ and label $\mathbf{Y} = \{\mathbf{y}_l\}_{l=1}^{L}$ from which the model can learn about the new classes, and an unlabeled data point $\mathbf{x}_*$ in the query dataset, the predictive probability of test label $y_* = k$ is:

$$p(y_* = k \mid \mathbf{x}_*, \mathbf{X}, \mathbf{Y}, \widehat{\mathbf{\Theta}}) = \int p(y_* = k \mid \mathbf{f}_*)\prod_{c=1}^{C}q(f_*^c \mid \mathbf{X}, \mathbf{Y}, \widehat{\mathbf{\Theta}})d\mathbf{f}_*, \quad \text{(4.6a)}$$

$$q(f_*^c \mid \mathbf{X}, \mathbf{Y}, \widehat{\mathbf{\Theta}}) = \int p(f_*^c \mid \mathbf{f}^c)q(\mathbf{f}^c \mid \mathbf{X}, \mathbf{Y}, \widehat{\mathbf{\Theta}})d\mathbf{f}^c = \mathcal{N}(f_*^c \mid \mu_*^c, \sigma_*^{2c}), \quad \text{(4.6b)}$$

where $k \in \{1, \ldots, C\}$, $\widehat{\mathbf{\Theta}}$ is the learned kernel hyperparameter in the training procedure, $p(y_* = k \mid \mathbf{f}_*)$ is the logistic-softmax likelihood, $\mathbf{f}_* = [f^1(\mathbf{x}_*), \ldots, f^C(\mathbf{x}_*)]^\top$, $q(\mathbf{f}^c \mid \mathbf{X}, \mathbf{Y}, \widehat{\mathbf{\Theta}})$ is Eq. (4.3d), $\mu_*^c = \mathbf{k}_{*l}^c\mathbf{K}_{ll}^{c^{-1}}\widetilde{\boldsymbol{\mu}}^c$ and $\sigma_*^{2c} = k_{**}^c - \mathbf{k}_{*l}^c\mathbf{K}_{ll}^{c^{-1}}\mathbf{k}_{l*}^c + \mathbf{k}_{*l}^c\mathbf{K}_{ll}^{c^{-1}}\widetilde{\mathbf{\Sigma}}^c\mathbf{K}_{ll}^{c^{-1}}\mathbf{k}_{l*}^c$. The integral in Eq. (4.6a) is intractable, so we resort to Monte Carlo to compute it and classify the test point by the highest predictive probability. The complete training and test procedure is described in Alg. 1 of Appendix VII.

# 5 Experiments

In this section, we present the results for few-shot classification tasks including accuracy and uncertainty quantification. We consider two challenging standard benchmark datasets: the Caltech-UCSD Birds [32] and mini-ImageNet [24].

## 5.1 Few-shot Classification and Domain Transfer

In this subsection, we describe the experimental setup and report the results of our few-shot classification tasks. While we notice that there are various settings for few-shot classification (e.g., the distribution calibration technique in Yang et al. [36]), we adopt the vanilla setting in Bayesian meta-learning to better compare the likelihoods. Following the procedure of prior work [20], we employ a standard Conv4 architecture [31] as the backbone and assess our models under six different settings, including 1-shot and 5-shot scenarios for both in-domain and cross-domain tasks. We benchmark our models against various baselines and state-of-the-art models, including Feature Transfer [5], Baseline++ [5], MatchingNet [31], ProtoNet [28], RelationNet [29], MAML [7], DKT [20], Bayesian MAML [38], ABML [23], LS [9], and OVE [27]. As for LS, we utilized the Gibbs sampling version implemented by Snell & Zemel [27], which is more computationally demanding than our mean-field method. We note that DKT, LS, and OVE are similar to our proposed method as they are all GP-based models but use different likelihood functions and inference methods. To ensure a fair comparison, we also apply the cosine kernel as in Patacchiola et al. [20] and Snell & Zemel [27].

We train and evaluate our models (denoted as CDKT) with the ELBO loss (denoted as **ML**) using the default number of epochs from Patacchiola et al. [20], and with the predictive likelihood loss (denoted as **PL**) using 800 epochs. However, we find that the ML ($\tau < 1$) domain-transfer experiment requires 800 epochs to avoid underfitting. We set the temperature parameter to $\tau = 0.5$ for the 5-shot PL experiment of mini-ImageNet and domain transfer, and $\tau = 0.2$ for all other experiments. Drawing inspiration from Section 3, we observe that placing a negative mean prior improves our method by 0.5% to 1% across all experiments. To ensure numerical stability, we train our models with a zero mean prior and use a constant negative mean of $-5$ at test time for all ML ($\tau < 1$) experiments. We also use this technique for the PL ($\tau < 1$) experiment of the domain-transfer task. We also note that the training of PL loss is generally less efficient than the ML loss, as it requires Monte Carlo sampling. Our mean field approximation converges at an extremely fast rate and we only need **2 steps** for task-level updates. More experimental details are provided in Appendix VIII.

We report the average accuracy and standard deviation of our models evaluated on 5 batches of 600 episodes with different random seeds in Table 1. Our model achieves the highest accuracy in both 1-shot (65.21%) and 5-shot (79.10%) experiments on the CUB dataset and achieves comparable results on the mini-ImageNet dataset. As for the domain-transfer task, we also achieve the highest result for the 1-shot (40.43%) scenario and a near-optimal result for the 5-shot (56.18%) scenario. Overall, while our mean-field approximation method works well for the ELBO loss (ML), the predictive likelihood loss (PL) generally does not match the performance demonstrated by the Gibbs sampling version of LS proposed by Snell & Zemel [27]. However, we find the PL loss effective for the domain-transfer scenario, indicating its potential at dealing with this type of task. Moreover, we observe a significant improvement (3% - 5%) in accuracy by adding a small temperature scaling ($\tau = 0.2$ or $0.5$) to the default logistic-softmax function ($\tau = 1$) in all scenarios. This finding suggests that our approach to controlling the *a priori* confidence of logistic-softmax is highly effective.

## 5.2 Uncertainty Quantification

Uncertainty quantification is an important aspect of few-shot learning because classifiers trained on limited data may have high variance and uncertainty in their predictions. This uncertainty can arise from a variety of sources, such as the small amount of training data, the complexity of the model, and the variability of the input data. A robust meta-learning method should be able to properly deal with such uncertainty, especially in high-risk areas including medical diagnosis, judicial adjudication, and autonomous driving.

We use 2 widely-applied metrics for uncertainty quantification, namely expected calibration error (ECE) [11] and maximum calibration error (MCE) [19]. ECE measures the average difference between confidence (probability outputs) and accuracy within each bin. MCE is similar to ECE but it

Table 1: Average 1-shot and 5-shot accuracy and standard deviation on 5-way few-shot classification. Baseline results are from Patacchiola et al. [20] and Snell & Zemel [27]. Results are evaluated over 5 batches of 600 episodes with different random seeds. We highlight the best results in bold.

| Method | CUB | | mini-ImageNet | | mini-ImageNet → CUB | |
|---|---|---|---|---|---|---|
| | 1-shot | 5-shot | 1-shot | 5-shot | 1-shot | 5-shot |
| Feature Transfer | $46.19 \pm 0.64$ | $68.40 \pm 0.79$ | $39.51 \pm 0.23$ | $60.51 \pm 0.55$ | $32.77 \pm 0.35$ | $50.34 \pm 0.27$ |
| Baseline++ | $61.75 \pm 0.95$ | $78.51 \pm 0.59$ | $47.15 \pm 0.49$ | $66.18 \pm 0.18$ | $39.19 \pm 0.12$ | $\mathbf{57.31 \pm 0.11}$ |
| MatchingNet | $60.19 \pm 1.02$ | $75.11 \pm 0.35$ | $48.25 \pm 0.65$ | $62.71 \pm 0.44$ | $36.98 \pm 0.06$ | $50.72 \pm 0.36$ |
| ProtoNet | $52.52 \pm 1.90$ | $75.93 \pm 0.46$ | $44.19 \pm 1.30$ | $64.07 \pm 0.65$ | $33.27 \pm 1.09$ | $52.16 \pm 0.17$ |
| RelationNet | $62.52 \pm 0.34$ | $78.22 \pm 0.07$ | $48.76 \pm 0.17$ | $64.20 \pm 0.28$ | $37.13 \pm 0.20$ | $51.76 \pm 1.48$ |
| MAML | $56.11 \pm 0.69$ | $74.84 \pm 0.62$ | $45.39 \pm 0.49$ | $61.58 \pm 0.53$ | $34.01 \pm 1.25$ | $48.83 \pm 0.62$ |
| DKT + Cosine | $63.37 \pm 0.19$ | $77.73 \pm 0.26$ | $48.64 \pm 0.45$ | $62.85 \pm 0.37$ | $40.22 \pm 0.54$ | $55.65 \pm 0.05$ |
| Bayesian MAML | $55.93 \pm 0.71$ | $72.87 \pm 0.26$ | $44.46 \pm 0.30$ | $62.60 \pm 0.25$ | $33.52 \pm 0.36$ | $51.35 \pm 0.16$ |
| Bayesian MAML (Chaser) | $53.93 \pm 0.72$ | $71.16 \pm 0.32$ | $43.74 \pm 0.46$ | $59.23 \pm 0.34$ | $36.22 \pm 0.50$ | $51.53 \pm 0.43$ |
| ABML | $49.57 \pm 0.42$ | $68.94 \pm 0.16$ | $37.65 \pm 0.22$ | $56.08 \pm 0.29$ | $29.35 \pm 0.26$ | $45.74 \pm 0.33$ |
| LS (Gibbs) + Cosine (ML) | $60.23 \pm 0.54$ | $74.58 \pm 0.25$ | $46.75 \pm 0.20$ | $59.93 \pm 0.31$ | $36.41 \pm 0.18$ | $50.33 \pm 0.13$ |
| LS (Gibbs) + Cosine (PL) | $60.07 \pm 0.29$ | $78.14 \pm 0.07$ | $47.05 \pm 0.20$ | $66.01 \pm 0.25$ | $36.73 \pm 0.26$ | $56.70 \pm 0.31$ |
| OVE PG GP + Cosine (ML) | $63.98 \pm 0.43$ | $77.44 \pm 0.18$ | $\mathbf{50.02 \pm 0.35}$ | $64.58 \pm 0.31$ | $39.66 \pm 0.18$ | $55.71 \pm 0.31$ |
| OVE PG GP + Cosine (PL) | $60.11 \pm 0.26$ | $79.07 \pm 0.05$ | $48.00 \pm 0.24$ | $\mathbf{67.14 \pm 0.23}$ | $37.49 \pm 0.11$ | $57.23 \pm 0.31$ |
| CDKT + Cosine (ML) ($\tau < 1$) | $\mathbf{65.21 \pm 0.45}$ | $\mathbf{79.10 \pm 0.33}$ | $47.54 \pm 0.21$ | $63.79 \pm 0.15$ | $\mathbf{40.43 \pm 0.43}$ | $55.72 \pm 0.45$ |
| CDKT + Cosine (ML) ($\tau = 1$) | $60.85 \pm 0.38$ | $75.98 \pm 0.33$ | $43.50 \pm 0.17$ | $59.69 \pm 0.20$ | $35.57 \pm 0.30$ | $52.42 \pm 0.50$ |
| CDKT + Cosine (PL) ($\tau < 1$) | $59.49 \pm 0.35$ | $76.95 \pm 0.28$ | $44.97 \pm 0.25$ | $60.87 \pm 0.24$ | $39.18 \pm 0.34$ | $56.18 \pm 0.28$ |
| CDKT + Cosine (PL) ($\tau = 1$) | $52.91 \pm 0.29$ | $73.34 \pm 0.40$ | $40.29 \pm 0.14$ | $60.23 \pm 0.16$ | $37.62 \pm 0.32$ | $54.32 \pm 0.19$ |

Table 2: Expected calibration error (ECE) and maximum calibration error (MCE) for 5-shot 5-way tasks on CUB, mini-ImageNet, and domain-transfer. Baseline results are from Snell & Zemel [27]. All metrics are computed on 3,000 random tasks from the test set.

| Method | CUB | | mini-ImageNet | | mini-ImageNet→CUB | |
|---|---|---|---|---|---|---|
| | ECE | MCE | ECE | MCE | ECE | MCE |
| Feature Transfer | 0.187 | 0.250 | 0.368 | 0.641 | 0.275 | 0.646 |
| Baseline++ | 0.421 | 0.502 | 0.395 | 0.598 | 0.315 | 0.537 |
| MatchingNet | 0.023 | 0.031 | 0.019 | 0.043 | 0.030 | 0.079 |
| ProtoNet | 0.034 | 0.059 | 0.035 | 0.050 | 0.009 | 0.025 |
| RelationNet | 0.438 | 0.593 | 0.330 | 0.596 | 0.234 | 0.554 |
| DKT + Cosine | 0.187 | 0.250 | 0.287 | 0.446 | 0.236 | 0.426 |
| Bayesian MAML | 0.018 | 0.047 | 0.027 | 0.049 | 0.048 | 0.077 |
| Bayesian MAML (Chaser) | 0.047 | 0.104 | 0.010 | 0.071 | 0.066 | 0.260 |
| LS (Gibbs) + Cosine (ML) | 0.371 | 0.478 | 0.277 | 0.490 | 0.220 | 0.513 |
| LS (Gibbs) + Cosine (PL) | 0.024 | 0.038 | 0.026 | 0.041 | 0.022 | 0.042 |
| OVE PG GP + Cosine (ML) | 0.026 | 0.043 | 0.026 | 0.039 | 0.049 | 0.066 |
| OVE PG GP + Cosine (PL) | **0.005** | **0.023** | **0.008** | 0.016 | 0.020 | 0.032 |
| CDKT + Cosine (ML) | **0.005** | 0.036 | 0.009 | **0.015** | **0.007** | **0.020** |
| CDKT + Cosine (PL) | 0.018 | 0.223 | 0.025 | 0.140 | 0.010 | 0.029 |

measures the maximum difference. Following the protocol of Patacchiola et al. [20] for evaluation, we first tune the temperature parameter on the validation set for alignment and then compute the ECE and MCE on the test set. The results of 5-shot experiments are summarized in Table 2. We observe that our introduction of temperature significantly improves the calibration of logistic-softmax compared to the one implemented by Snell & Zemel [27]. Specifically, we obtain the lowest ECE on CUB (0.005) and domain transfer (0.007) and the lowest MCE on mini-ImageNet (0.015) and domain transfer (0.020). Our model performs marginally worse than state-of-the-art in MCE on CUB and ECE on mini-ImageNet. Overall, this result indicates that our model is highly reliable on uncertainty calibration, demonstrating promising robustness in few-shot scenarios.

Figure 3 shows the reliability diagrams of our models. The confidence barplot should match the diagonal line for a robust uncertainty quantification model. We observe that the models trained with the ELBO loss (ML) generally perform better than those trained with the predictive likelihood (PL). Specifically, ML models closely match the diagonal line while PL models tend to underestimate the accuracy of low-confidence outputs. In conclusion, our findings indicate that ML models are better at handling uncertainty in the context of mean-field approximation inference. This is an interesting result as it contrasts the findings in Snell & Zemel [27], where PL models with Gibbs sampling generally perform better. Nonetheless, we leave the analysis of this phenomenon to future work.

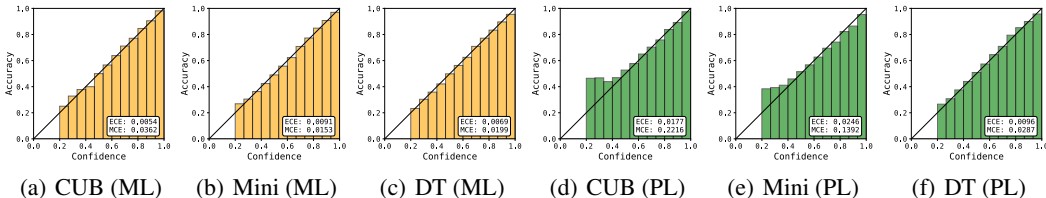

| (a) CUB (ML) | (b) Mini (ML) | (c) DT (ML) | (d) CUB (PL) | (e) Mini (PL) | (f) DT (PL) |

Figure 3: Reliability diagrams on 5-shot classification with expected calibration error (ECE) and maximum calibration error (MCE) metrics. Mini denotes the mini-ImageNet dataset, and DT denotes the domain transfer task of CUB → mini-ImageNet. Results are computed on 3,000 test tasks.

## 6    Related Work

GP classification has been extensively studied, and several approaches have been proposed to address its challenges, including label regression and Laplace approximation [25; 34]. Recently, Polson et al. [21] introduced Polya-Gamma augmentation, and Galy-Fajou et al. [9]; Snell & Zemel [27] utilized different likelihood functions and data augmentation to approximate the softmax function. These methods have advanced the field by providing intuitive frameworks and improving modeling capabilities for GP classifiers. Our work builds upon these foundations by focusing on a more generalized likelihood formulation with enhanced flexibility and modeling capabilities for GP classifiers.

Bayesian meta-learning has been explored through various approaches to leverage prior knowledge and adapt to new tasks. Finn et al. [8] introduced a Bayesian hierarchical modeling perspective, capturing uncertainty at different levels. Grant et al. [10] recast meta-learning as inference in a GP. Yoon et al. [38] introduced Bayesian MAML on the basis of Finn et al. [7]. Patacchiola et al. [20]; Snell & Zemel [27] utilized GPs with deep kernels for task-specific inference. These works made contributions to Bayesian meta-learning by addressing parameter updates, uncertainty modeling, and prior distributions. Our work contributes by providing an effective alternative for task-level updates and further provides insights into the coordination problem of bi-level optimization.

## 7    Limitations

While the logistic-softmax with temperature has proven to be theoretically superior to softmax in data modeling, its performance and suitability are only verified in Bayesian meta-learning, leading to certain limitations. Further research is necessary to explore the logistic-softmax function's performance and adaptability to various domains and problem settings.

## 8    Conclusions

In this paper, we introduce the logistic-softmax function with temperature which is simple yet highly effective in improving classification accuracy and uncertainty calibration. Furthermore, we delve into the theoretical property of the redesigned logistic-softmax function including both limiting behavior and data modeling capability. Moreover, we apply mean-field approximation for deep kernel based GP meta-learning for the first time. We also shed some light on the coordination problem between the inner loop and the outer loop that appeared in bi-level optimization. In the future, it is an interesting track to apply redesigned logistic-softmax to other domains such as multi-label classification.

## Acknowledgments and Disclosure of Funding

This work was supported by NSFC Project (No. 62106121), the MOE Project of Key Research Institute of Humanities and Social Sciences (22JJD110001), the fund for building world-class universities (disciplines) of Renmin University of China (No. KYGJC2023012), and the Public Computing Cloud, Renmin University of China.

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
