# Appendix

## I  Proof of Theorem 3.1

*Proof.* We assume that $f_n^c \neq 0$ for all $c = 1, \ldots, C$. For a given index $k$, if $f_n^k > 0$, $\sigma(f_n^k/\tau)$ goes to 1 as $\tau \to 0$. Otherwise, $\sigma(f_n^k/\tau)$ goes to 0 if $f_n^k < 0$. Since $\mathrm{LS}(\mathbf{f}, \tau)_k = \frac{\sigma(f_n^k/\tau)}{\sum_{c=1}^C \sigma(f_n^c/\tau)}$, when $\max_{c=1,\ldots,C} f_n^c > 0$, we get the result combining these observations. As for $\max_{c=1,\ldots,C} f_n^c < 0$, we have

$$
\begin{aligned}
\lim_{\tau \to 0^+} \mathrm{LS}(\mathbf{f}, \tau)_k &= \left( 1 + \lim_{\tau \to 0^+} \sum_{c \neq k} \frac{1 + \exp(-f_n^k/\tau)}{1 + \exp(-f_n^c/\tau)} \right)^{-1} \\
&= \left( 1 + \lim_{\tau \to 0^+} \sum_{c \neq k} \frac{f_n^k \exp(-f_n^k/\tau)/\tau^2}{f_n^c \exp(-f_n^c/\tau)/\tau^2} \right)^{-1} \\
&= \left( 1 + \lim_{\tau \to 0^+} \sum_{c \neq k} \frac{f_n^k}{f_n^c} \exp(-(f_n^k - f_n^c)/\tau) \right)^{-1},
\end{aligned}
$$

where we use L'Hôpital's rule in the second equality. Since $f_n^k/f_n^c > 0$, $\exp(-(f_n^k - f_n^c)/\tau)$ goes to 0 if $f_n^k > f_n^c$ and goes to $+\infty$ otherwise. Thus, we have $\lim_{\tau \to 0^+} \mathrm{LS}(\mathbf{f}_n, \tau)_k = \mathbb{I}\{k = c^*\}$, where $\mathbb{I}\{\cdot\}$ is the indicator function. $\qquad \square$

## II  Proof of Theorem 3.2

*Proof.* Without the loss of generality, we use $\tau = 1$ in the following proof. Notice that for logistic-softmax, we have

$$
p(y = k|\mathbf{f}_n - C') = \frac{1}{\sum_{c=1}^C \sigma(f_n^c - C')/\sigma(f_n^k - C')}, C' \in \mathbb{R}.
$$

It's sufficient to prove that the denominator converges to that of softmax at each point $\mathbf{f}_n$ as $C'$ goes to infinity. This is true since for all $c, k \in [C]$ we have

$$
\begin{aligned}
\frac{\sigma(f_n^c - C')}{\sigma(f_n^k - C')} &= \exp(f_n^c - f_n^k) \cdot \frac{1 + \exp(f_n^k - C')}{1 + \exp(f_n^c - C')} \\
&\to \exp(f_n^c - f_n^k)
\end{aligned}
$$

when $C' \to \infty$.

For the claim that $S(\mathbf{f}_n - C_0) = S(\mathbf{f}_n)$, one only needs to observe that the likelihood of softmax can be rewritten as

$$
\begin{aligned}
p(y_n = k|\mathbf{f_n}) &= \frac{1}{1 + \sum_{j \neq k} \exp(f_n^j - f_n^k)} \\
&= \frac{1}{1 + \sum_{j \neq k} \exp((f_n^j - C_0) - (f_n^k - C_0))}.
\end{aligned}
$$

We have shown that softmax is translational invariant w.r.t. its input vector $\mathbf{f_n}$, therefore completing the proof. $\qquad \square$

## III  Proof of Theorem 3.3

*Proof.* Without the loss of generality, we use $\tau = 1$ in the following proof.

To begin with, we prove the first equation and then give the proof of the second part of Theorem 3.3. We introduce some extra notations that are used throughout the proof. Denote $\mathbf{f}^c = (f_1^c, \ldots, f_N^c)^\top \in \mathbb{R}^N$ as the logits of N given points for class $c$. We write $\mathbf{F} = (f_n^c)_{N \times C} \in \mathbb{R}^{N \times C}$ and $\mathbf{f} = vec(\mathbf{F})$ as the logit vector, where we stack the logits of each class. It's straightforward to verify that $\mathbf{f} \sim \mathcal{N}\left(\mathrm{vec}(\mathbf{a}\mathbf{1}_C^T), \mathbf{K}\right)$, where $\mathbf{a}$ is the mean vector on given points, $\mathbf{K} = \mathrm{diag}(\mathbf{K}^1, \ldots, \mathbf{K}^C) \in$

$\mathbb{R}^{NC \times NC}$ is the block diagonal matrix and $\mathbf{K}^c$ is the kernel matrix for each class. Denote $\mathbf{y} \in \mathbf{R}^N$ as the label vector for the $N$ given points, where $y_n \in [C]$.

For the first equation, notice that

$$p(\mathbf{y}) = \int \prod_{n=1}^N \frac{1}{1 + \sum_{j \neq y_n} \exp(f_n^j - f_n^{y_n})} p(\mathbf{f}) d\mathbf{f}.$$

Denote $\widetilde{\mathbf{f}} \in \mathbb{R}^{NC}$ as follows:

$$\widetilde{f}_n^j = \begin{cases} f_n^j - f_n^{y_n}, & \text{if } j \neq y_n \\ f_n^{y_n}, & \text{if } j = y_n. \end{cases} \tag{1}$$

We denote $\widetilde{\mathbf{f}}_y \in \mathbb{R}^N$ where the $n$-th element of this vector equals to $f_n^{y_n}$. We also write $\widetilde{\mathbf{f}}_{-y} \in \mathbb{R}^{N(C-1)}$ to denote the rest of the elements in $\widetilde{\mathbf{f}}$. Since Eq. (1) is a linear transformation of $\mathbf{f}$, it's straightforward to verify that $\widetilde{\mathbf{f}}_{-y}$ is a multivariate Gaussian variable with zero mean, thus the distribution of it is irrelevant to $a$. Then we use the substitution rule for definite integrals and derive

$$\begin{aligned} p(\mathbf{y}) &= \int \prod_{n=1}^N \frac{1}{1 + \sum_{j \neq y_n} \exp(\widetilde{f}_n^j)} p(\widetilde{\mathbf{f}}_y) d\widetilde{\mathbf{f}}_y \\ &= \int \prod_{n=1}^N \frac{1}{1 + \sum_{j \neq y_n} \exp(\widetilde{f}_n^j)} p(\widetilde{\mathbf{f}}_y \mid \widetilde{\mathbf{f}}_{-y}) p(\widetilde{\mathbf{f}}_{-y}) d\widetilde{\mathbf{f}} \\ &= \int \prod_{n=1}^N \frac{1}{1 + \sum_{j \neq y_n} \exp(\widetilde{f}_n^j)} p(\widetilde{\mathbf{f}}_{-y}) d\widetilde{\mathbf{f}}_{-y}, \end{aligned}$$

where $p(\widetilde{\mathbf{f}}_y \mid \widetilde{\mathbf{f}}_{-y})$ is integrated out in the third equation. Thus, $p(\mathbf{y})$ is irrelevant to $a$ since the distribution of $\widetilde{\mathbf{f}}_{-y}$ is irrelevant to $a$. Therefore, we complete the proof of the first equation by showing that $p(\mathbf{y})$ only depends on $k^c$.

Now we give proof to the second part. First we denote the marginal likelihood of $\mathbf{y}$ induced by ls and softmax likelihood with gaussian prior mean $\mathbf{a}$ and covariance $\mathbf{K}$ as $p_{ls}(\mathbf{y}|\mathbf{a}, \mathbf{K})$ and $p_s(\mathbf{y}|\mathbf{K})$ respectively. We start by pointing out the desired convergence result as follows:

$$\begin{aligned} \lim_{\mathbf{a} \to -\infty} p_{ls}(\mathbf{y}|\mathbf{a}, \mathbf{K}) &= \lim_{\mathbf{a} \to -\infty} \int p_{ls}(\mathbf{y}|\mathbf{F}) p(\mathbf{F}|\mathbf{a}, \mathbf{K}) d\mathbf{F} \\ &= \lim_{\mathbf{a} \to \infty} \int p_{ls}(\mathbf{y}|\mathbf{F} - \mathbf{a}\mathbf{1}_C^T) p(\mathbf{F}|\mathbf{0}, \mathbf{K}) d\mathbf{F} \\ &= \int \lim_{\mathbf{a} \to \infty} p_{ls}(\mathbf{y}|\mathbf{F} - \mathbf{a}\mathbf{1}_C^T) p(\mathbf{F}|\mathbf{0}, \mathbf{K}) d\mathbf{F} \\ &= \int p_s(\mathbf{y}|\mathbf{F}) p(\mathbf{F}|\mathbf{0}, \mathbf{K}) d\mathbf{F} = p_s(\mathbf{y}|\mathbf{K}), \end{aligned} \tag{2}$$

where the second equation holds due to the property of multivariate Gaussian variable. In the third equation, we need to interchange the integration and limiting operations. To guarantee its feasibility, we rely on the Dominated convergence theorem (DCT). To verify the condition of DCT, notice that,

$$\begin{aligned} p_{ls}(\mathbf{y}|\mathbf{F} - \mathbf{a}\mathbf{1}_C^T) p(\mathbf{F}|\mathbf{0}, \mathbf{K}) &= \prod_{n=1}^N \frac{\sigma(f_n^{y_n} - a_n)}{\sum_{j=1}^C \sigma(f_n^j - a_n)} p(\mathbf{F}|\mathbf{0}, \mathbf{K}) \\ &\leq p(\mathbf{F}|\mathbf{0}, \mathbf{K}). \end{aligned}$$

This implies that $p_{ls}(\mathbf{y}|\mathbf{F} - \mathbf{a}\mathbf{1}_C^T) p(\mathbf{F}|\mathbf{0}, \mathbf{K})$ is dominated by the Gaussian prior $p(\mathbf{F}|\mathbf{0}, \mathbf{K})$, which is integrable. Since Theorem 3.2 directly implies

$$\lim_{\mathbf{a} \to \infty} p_{ls}(\mathbf{y}|\mathbf{F} - \mathbf{a}\mathbf{1}_C^T) = p_s(\mathbf{y}|\mathbf{F}),$$

thus by DCT, the desired convergence result in Eq. (2) is proved.

Our next step is to define a suitable mean and kernel function class for $a$ and $k^c$ respectively. For simplicity, we consider each sample point $x_i \in \mathbb{R}^p$. Define

$$\mathscr{A} := \{f : \mathbb{R}^p \to \overline{\mathbb{R}}\},$$
$$\mathscr{K} := \{f : \mathbb{R}^p \times \mathbb{R}^p \to \mathbb{R}, f \text{ is postive semi-definite}\}.$$

We also say $a_0(\mathbf{x}) \equiv -\infty, \forall \mathbf{x} \in \mathbb{R}^p$, where $a_0 \in \mathscr{A}$. We define the marginalized likelihood of $\mathbf{y}$ induced by the logistic-softmax likelihood with $a_0$ and $k^c$ evaluated at $\mathbf{X}$ as,

$$
\begin{aligned}
p_{ls}(\mathbf{y}|\mathbf{X}, a_0, k^c) &:= \lim_{\mathbf{a} \to -\infty} p_{ls}(\mathbf{y}|\mathbf{a}, \mathbf{K}) \\
&= p_s(\mathbf{y}|\mathbf{K}),
\end{aligned}
\tag{3}
$$

where the second equation is from Eq. (2). Finally, we define $\mathscr{F}(\text{LS} \mid \mathscr{A}, \mathscr{K})$ and $\mathscr{F}(\text{S} \mid \mathscr{K})$as,

$$\mathscr{F}(\text{LS} \mid \mathscr{A}, \mathscr{K}) := \{f : f(\mathbf{y}) = p_{ls}(\mathbf{y}|\mathbf{a}, \mathbf{K}), a_i = a(\mathbf{x_i}), k_{ij}^c = k^c(\mathbf{x}_i, \mathbf{x}_j), a \in \mathscr{A}, k^c \in \mathscr{K}, \mathbf{X} \in \mathbb{R}^{N \times p}\},$$
$$\mathscr{F}(\text{S} \mid \mathscr{K}) := \{f : f(\mathbf{y}) = p_s(\mathbf{y}|\mathbf{K}), k_{ij}^c = k^c(\mathbf{x}_i, \mathbf{x}_j), k^c \in \mathscr{K}, X \in \mathbb{R}^{N \times p}\}.$$

For each $p_s(\cdot|\mathbf{X}, k^c)$ in $\mathscr{F}(\text{S} \mid \mathscr{K})$, we have $p_{ls}(\cdot|\mathbf{X}, a_0, k^c) = p_s(\cdot|\mathbf{X}, k^c)$ using Eq. (3), where $p_{ls}(\cdot|\mathbf{X}, a_0, k^c) \in \mathscr{F}(\text{LS} \mid \mathscr{A}, \mathscr{K})$. Thus, we have proved that

$$\mathscr{F}(\text{S} \mid \mathscr{K}) \subset \mathscr{F}(\text{LS} \mid \mathscr{A}, \mathscr{K}).$$

$\square$

## IV   Derivation of Augmented Joint Distribution

In this section, we derive the Gibbs sampler and mean-field variational inference for a specific task. The usual likelihood function for multiclass classification is the softmax function. Here, we replace the softmax function with the logistic-softmax function [9]

$$p(y_n = k \mid \mathbf{f}_n) = \frac{\sigma(f_n^k/\tau)}{\sum_{c=1}^C \sigma(f_n^c/\tau)},
\tag{4}$$

where $f_n^c = f^c(\mathbf{x}_n)$, $\mathbf{f}_n = [f_n^1, \ldots, f_n^C]^\top$, $k \in \{1, \ldots, C\}$ and we omit the conditioning on $\mathbf{x}_n$. In the following, we augment three auxiliary latent variables to make the likelihood appear in a conjugate form.

**Augmentation of Gamma Variables**   We utilize the integral identity $1/z = \int_0^\infty \exp(-\lambda z)d\lambda$ to express Eq. (4)

$$
\begin{aligned}
p(\mathbf{y} \mid \mathbf{F}) &= \prod_{n=1}^N \frac{\sigma(f_n^k/\tau)}{\sum_{c=1}^C \sigma(f_n^c/\tau)} = \prod_{n=1}^N \sigma(f_n^k/\tau) \int_0^\infty \exp(-\lambda_n \sum_{c=1}^C \sigma(f_n^c/\tau))d\lambda_n \\
&= \int_0^\infty \cdots \int_0^\infty \prod_{n=1}^N \sigma(f_n^k/\tau) \exp(-\lambda_n \sum_{c=1}^C \sigma(f_n^c/\tau))d\lambda_1 \cdots d\lambda_N,
\end{aligned}
$$

where $\mathbf{y} = [y_1, \ldots, y_N]^\top$, $\mathbf{F}$ is the $N \times C$ matrix of $f_n^c$. Therefore, we obtain the augmented likelihood of Gamma variables

$$p(\mathbf{y}, \boldsymbol{\lambda} \mid \mathbf{F}) = \prod_{n=1}^N \sigma(f_n^k/\tau) \prod_{c=1}^C \exp(-\lambda_n \sigma(f_n^c/\tau)),
\tag{5}$$

where $\boldsymbol{\lambda} = [\lambda_1, \ldots, \lambda_N]^\top$.

**Augmentation of Poisson Variables**   We rewrite the exponential term in Eq. (5) using the moment generating function of the Poisson distribution $\exp(\lambda(z-1)) = \sum_{m=0}^\infty z^m \text{Po}(m \mid \lambda)$ and the logistic symmetry property $\sigma(x) = 1 - \sigma(-x)$.

$$p(\mathbf{y}, \boldsymbol{\lambda} \mid \mathbf{F}) = \prod_{n=1}^N \sigma(f_n^k/\tau) \prod_{c=1}^C \sum_{m_n^c=0}^\infty \sigma(-f_n^c/\tau)^{m_n^c} \text{Po}(m_n^c|\lambda_n).$$

Therefore, we obtain the augmented likelihood of Poisson variables

$$p(\mathbf{y}, \boldsymbol{\lambda}, \mathbf{M} \mid \mathbf{F}) = \prod_{n=1}^{N} \sigma(f_n^k/\tau) \prod_{c=1}^{C} \sigma(-f_n^c/\tau)^{m_n^c} \mathrm{Po}(m_n^c|\lambda_n), \tag{6}$$

where $\mathbf{M}$ is the $N \times C$ matrix of $m_n^c$.

**Augmentation of Pólya-Gamma Variables**    The logistic function in Eq. (6) can be rewritten as a scale mixture of Gaussians utilizing the Pólya-Gamma representation [21]

$$\sigma(z) = 2^{-1} e^{z/2} \int_0^\infty e^{-\omega z^2/2} \mathrm{PG}(\omega \mid 1, 0) d\omega,$$

where $\mathrm{PG}(\omega \mid 1, 0)$ is the Pólya-Gamma distribution.

$$p(\mathbf{Y}, \boldsymbol{\lambda}, \mathbf{M} \mid \mathbf{F}) = \int_0^\infty \cdots \int_0^\infty \prod_{n=1}^{N} \prod_{c=1}^{C} 2^{-(y_n^c + m_n^c)} \exp\left( \frac{y_n^c - m_n^c}{2} \frac{f_n^c}{\tau} - \frac{\omega_n^c}{2} \left(\frac{f_n^c}{\tau}\right)^2 \right)$$

$$\mathrm{PG}(\omega_n^c \mid m_n^c + y_n^c, 0) \frac{\lambda_n^{m_n^c}}{m_n^c!} \exp(-\lambda_n) d\omega_1^1 \cdots d\omega_N^C,$$

where we rewrite $\mathbf{y}$ in the one-hot encoding form $\mathbf{Y}$ which is a $N \times C$ matrix. Therefore, we obtain the augmented likelihood of Pólya-Gamma variables

$$p(\mathbf{Y}, \boldsymbol{\lambda}, \mathbf{M}, \boldsymbol{\Omega} \mid \mathbf{F}) = \prod_{n=1}^{N} \prod_{c=1}^{C} 2^{-(y_n^c + m_n^c)} \exp\left( \frac{y_n^c - m_n^c}{2} \frac{f_n^c}{\tau} - \frac{\omega_n^c}{2} \left(\frac{f_n^c}{\tau}\right)^2 \right) \mathrm{PG}(\omega_n^c \mid m_n^c + y_n^c, 0)$$

$$\frac{\lambda_n^{m_n^c}}{m_n^c!} \exp(-\lambda_n), \tag{7}$$

where $\boldsymbol{\Omega}$ is the $N \times C$ matrix of $\omega_n^c$.

**Augmented Joint Distribution**    Introducing the GP priors on $f^c$, we obtain the augmented joint distribution

$$p(\mathbf{Y}, \boldsymbol{\lambda}, \mathbf{M}, \boldsymbol{\Omega}, \mathbf{F}) = \prod_{n=1}^{N} \prod_{c=1}^{C} 2^{-(y_n^c + m_n^c)} \exp\left( \frac{y_n^c - m_n^c}{2} \frac{f_n^c}{\tau} - \frac{\omega_n^c}{2} \left(\frac{f_n^c}{\tau}\right)^2 \right) \mathrm{PG}(\omega_n^c \mid m_n^c + y_n^c, 0)$$

$$\frac{\lambda_n^{m_n^c}}{m_n^c!} \exp(-\lambda_n) \cdot \prod_{c=1}^{C} \mathcal{N}(\mathbf{f}^c \mid \mathbf{a}^c, \mathbf{K}^c), \tag{8}$$

where $\mathbf{f}^c = [f_1^c, \ldots, f_N^c]^\top$ is the $c$-th column of $\mathbf{F}$, $\mathbf{a}^c$ is the mean and $\mathbf{K}^c$ is the kernel matrix w.r.t. observations for $c$-th class.

## V    Mean-field Variational Inference

The aforementioned Gibbs sampler is efficient because of closed-form solutions, but it is still not efficient enough because the sampling from a Pólya-Gamma distribution is time-consuming. In order to improve efficiency, the mean-field variational inference algorithm is proposed. In the mean-field algorithm, we need to approximate the true posterior $p(\boldsymbol{\lambda}, \mathbf{M}, \boldsymbol{\Omega}, \mathbf{F} \mid \mathbf{Y})$ by a variational distribution which is assumed to factorize over some partition of latent variables. Here, we assume the variational distribution $q(\boldsymbol{\lambda}, \mathbf{M}, \boldsymbol{\Omega}, \mathbf{F}) = q_1(\mathbf{M}, \boldsymbol{\Omega}) q_2(\boldsymbol{\lambda}, \mathbf{F})$. Following the traditional mean-field method [3], the optimal distribution for each factor can be expressed as

$$\log q_1(\mathbf{M}, \boldsymbol{\Omega}) = \mathbb{E}_{q_2} \log p(\mathbf{Y}, \boldsymbol{\lambda}, \mathbf{M}, \boldsymbol{\Omega}, \mathbf{F}) + C_1,$$
$$\log q_2(\boldsymbol{\lambda}, \mathbf{F}) = \mathbb{E}_{q_1} \log p(\mathbf{Y}, \boldsymbol{\lambda}, \mathbf{M}, \boldsymbol{\Omega}, \mathbf{F}) + C_2,$$

where $C_1$ and $C_2$ are constants. Substituting Eq. (8), we obtain

$$q_1(\mathbf{\Omega}|\mathbf{M}) = \prod_{n,c=1}^{N,C} \mathrm{PG}(\omega_n^c \mid m_n^c + y_n^c, \widetilde{f}_n^c), \quad (9a)$$

$$q_2(\mathbf{\lambda}) = \prod_{n=1}^{N} \mathrm{Ga}(\lambda_n \mid \alpha_n, C), \quad (9c)$$

$$q_1(\mathbf{M}) = \prod_{n,c=1}^{N,C} \mathrm{Po}(m_n^c \mid \gamma_n^c), \quad (9b)$$

$$q_2(\mathbf{F}) = \prod_{c=1}^{C} \mathcal{N}(\mathbf{f}^c \mid \widetilde{\boldsymbol{\mu}}^c, \widetilde{\boldsymbol{\Sigma}}^c), \quad (9d)$$

where

$$\widetilde{f}_n^c = \frac{1}{\tau}\sqrt{\mathbb{E}[f_n^{c2}]} = \frac{1}{\tau}\sqrt{\widetilde{\mu}_n^{c2} + \widetilde{\sigma}_{nn}^{c2}}, \quad (10a)$$

$$\widetilde{\boldsymbol{\Sigma}}^c = (\mathrm{diag}(\bar{\omega}_n^c/\tau^2) + \mathbf{K}^{c^{-1}})^{-1}, \quad (10d)$$

$$\gamma_n^c = \frac{\exp(\psi(\alpha_n) - \widetilde{\mu}_n^c/2\tau)}{2C \cosh(\widetilde{f}_n^c/2)}, \quad (10b)$$

$$\widetilde{\boldsymbol{\mu}}^c = \frac{1}{2\tau}\widetilde{\boldsymbol{\Sigma}}^c(\mathbf{y}^c - \boldsymbol{\gamma}^c) + \widetilde{\boldsymbol{\Sigma}}^c\mathbf{K}^{c^{-1}}\mathbf{a}^c, \quad (10e)$$

$$\alpha_n = \sum_{c=1}^{C} \gamma_n^c + 1, \quad (10c)$$

$$\bar{\omega}_n^c = \mathbb{E}[\omega_n^c] = \frac{\gamma_n^c + y_n^c}{2\widetilde{f}_n^c} \tanh\frac{\widetilde{f}_n^c}{2}. \quad (10f)$$

## VI  ELBO and Derivative

In this section, we derived the evidence lower bound for a specific task which is used to be optimized w.r.t. the hyperparameters of deep kernels [9]:

$$\log p(\mathbf{Y}) \geq \mathcal{L} = \mathbb{E}_q[\log p(\mathbf{Y} \mid \boldsymbol{\lambda}, \mathbf{M}, \boldsymbol{\Omega}, \mathbf{F})] - \mathrm{KL}(q(\boldsymbol{\lambda}, \mathbf{M}, \boldsymbol{\Omega}, \mathbf{F})||p(\boldsymbol{\lambda}, \mathbf{M}, \boldsymbol{\Omega}, \mathbf{F})), \quad (11)$$

where we omit the conditioning on hyperparameters $\boldsymbol{\Theta}$,

$$\mathbb{E}_q[\log p(\mathbf{Y} \mid \boldsymbol{\lambda}, \mathbf{M}, \boldsymbol{\Omega}, \mathbf{F})] = \sum_{n=1,c=1}^{N,C} -(y_n^c + \gamma_n^c)\log 2 + \frac{y_n^c - \gamma_n^c}{2\tau}\widetilde{\mu}_n^c - \frac{\bar{\omega}_n^c}{2}\widetilde{f}_n^{c2}, \quad (12a)$$

$$\mathrm{KL}(q(\boldsymbol{\lambda}, \mathbf{M}, \boldsymbol{\Omega}, \mathbf{F})||p(\boldsymbol{\lambda}, \mathbf{M}, \boldsymbol{\Omega}, \mathbf{F})) = \mathrm{KL}(q(\mathbf{F})||p(\mathbf{F})) + \mathrm{KL}(q(\boldsymbol{\lambda}, \mathbf{M}, \boldsymbol{\Omega})||p(\boldsymbol{\lambda}, \mathbf{M}, \boldsymbol{\Omega})), \quad (12b)$$

$$\mathrm{KL}(q(\mathbf{F})||p(\mathbf{F})) = \frac{1}{2}\sum_{c=1}^{C}(\log|\mathbf{K}^c| - \log|\widetilde{\boldsymbol{\Sigma}}^c| - N + \mathrm{Tr}[\mathbf{K}^{c^{-1}}\widetilde{\boldsymbol{\Sigma}}^c] + (\mathbf{a}^c - \widetilde{\boldsymbol{\mu}}^c)^\top\mathbf{K}^{c^{-1}}(\mathbf{a}^c - \widetilde{\boldsymbol{\mu}}^c)),$$

$$(12c)$$

$$\mathrm{KL}(q(\boldsymbol{\lambda}, \mathbf{M}, \boldsymbol{\Omega})||p(\boldsymbol{\lambda}, \mathbf{M}, \boldsymbol{\Omega})) = \mathrm{KL}(q(\boldsymbol{\lambda})||p(\boldsymbol{\lambda})) + \mathbb{E}_{q(\boldsymbol{\lambda})}[\mathrm{KL}(q(\mathbf{M})||p(\mathbf{M} \mid \boldsymbol{\lambda}))] \quad (12d)$$
$$+ \mathbb{E}_{q(\mathbf{M})}[\mathrm{KL}(q(\boldsymbol{\Omega} \mid \mathbf{M})||p(\boldsymbol{\Omega} \mid \mathbf{M}))],$$

$$\mathrm{KL}(q(\boldsymbol{\lambda})||p(\boldsymbol{\lambda})) = \sum_{n=1}^{N} -\alpha_n + \log C - \log\Gamma(\alpha_n) - (1 - \alpha_n)\psi(\alpha_n), \quad (12e)$$

$$\mathbb{E}_{q(\boldsymbol{\lambda})}[\mathrm{KL}(q(\mathbf{M})||p(\mathbf{M} \mid \boldsymbol{\lambda}))] = \sum_{n=1,c=1}^{N,C} \gamma_n^c(\log\gamma_n^c - 1) - \gamma_n^c(\psi(\alpha_n) - \log C) + \frac{\alpha_n}{C}, \quad (12f)$$

$$\mathbb{E}_{q(\mathbf{M})}[\mathrm{KL}(q(\boldsymbol{\Omega} \mid \mathbf{M})||p(\boldsymbol{\Omega} \mid \mathbf{M}))] = \sum_{n=1,c=1}^{N,C} -\frac{\widetilde{f}_n^{c2}}{2}\bar{\omega}_n^c + (\gamma_n^c + y_n^c)\log\cosh(\frac{\widetilde{f}_n^c}{2}). \quad (12g)$$

We can get the analytical ELBO by summing up Eqs. (12a), (12c) and (12e) to (12g).

$$
\begin{aligned}
\mathcal{L} = \sum_{n=1,c=1}^{N,C} & -(y_n^c + \gamma_n^c)\log 2 + \frac{y_n^c - \gamma_n^c}{2\tau}\widetilde{\mu}_n^c - \frac{\bar{\omega}_n^c}{2}\widetilde{f}_n^{c^2} \\
& - \frac{1}{2}\sum_{c=1}^{C}(\log|\mathbf{K}^c| - \log|\widetilde{\mathbf{\Sigma}}^c| - N + \mathrm{Tr}[\mathbf{K}^{c^{-1}}\widetilde{\mathbf{\Sigma}}^c] + (\mathbf{a}^c - \widetilde{\boldsymbol{\mu}}^c)^\top \mathbf{K}^{c^{-1}}(\mathbf{a}^c - \widetilde{\boldsymbol{\mu}}^c)) \\
& - \sum_{n=1}^{N} -\alpha_n + \log C - \log\Gamma(\alpha_n) - (1-\alpha_n)\psi(\alpha_n) \\
& - \sum_{n=1,c=1}^{N,C} \gamma_n^c(\log\gamma_n^c - 1) - \gamma_n^c(\psi(\alpha_n) - \log C) + \frac{\alpha_n}{C} \\
& - \sum_{n=1,c=1}^{N,C} -\frac{\widetilde{f}_n^{c^2}}{2}\bar{\omega}_n^c + (\gamma_n^c + y_n^c)\log\cosh(\frac{\widetilde{f}_n^c}{2}).
\end{aligned}
\tag{13}
$$

The gradient of ELBO w.r.t. $\boldsymbol{\Theta}$ can be computed by the automatic differentiation technique.

## VII    Algorithm

---

**Algorithm 1:** Efficient Bayesian Meta-learning for Few-shot Classification

---

**Training:**
**Input:** Support and query data $\{\mathbf{X}_t\}_{t=1}^T$, $\{\mathbf{Y}_t\}_{t=1}^T$ for $T$ tasks
**Output:** Hyperparameters $\boldsymbol{\Theta}$ for the kernels
Initialize the variational parameters of each task and hyperparameters of the kernels;
**for** *Iteration* **do**
 # All tasks are implemented in parallel
 **for** *Task t* **do**
  # Update task-level variational parameters until convergence
  **for** *Step* **do**
   Update $\widetilde{f}_n^c, \gamma_n^c, \alpha_n, \widetilde{\mathbf{\Sigma}}^c, \widetilde{\boldsymbol{\mu}}^c, \bar{\omega}_n^c$ iteratively by Eq. (4.4a) − (Eq. (4.4f))
  **end**
 **end**
 # Update meta-level hyperparameters
 Update $\boldsymbol{\Theta}$ by $\nabla_{\boldsymbol{\Theta}}\mathcal{L}$
**end**

**Test:**
**Input:** Support data $\mathbf{X}, \mathbf{Y}$; query data $\mathbf{x}_*$; learned hyperparameters $\widehat{\boldsymbol{\Theta}}$
**Output:** Label $y_*$
Initialize the variational parameters of test task;
**for** *Iteration* **do**
 # Update test-task variational parameters until convergence
 **for** *Step* **do**
  Update $\widetilde{f}_n^c, \gamma_n^c, \alpha_n, \widetilde{\mathbf{\Sigma}}^c, \widetilde{\boldsymbol{\mu}}^c, \bar{\omega}_n^c$ iteratively by Eq. (4.4a) − (Eq. (4.4f))
 **end**
**end**
# Predict the test label
Predict $y_*$ by Eq. (4.6).

---

## VIII    Experimental Details

**Datasets**

We use three dataset scenarios as described below.

1. **CUB.** There are 200 classes and 11788 images in the Caltech-UCSD Birds (CUB) dataset. We use the common split of 100 training, 50 validation, and 50 test classes [27].

2. **mini-ImageNet.** There are 100 classes associated with 600 images for each class in the mini-ImageNet dataset. We also use the usual split of 64 training, 16 validation, and 20 test classes as applied in Snell & Zemel [27] .

3. **mini-ImageNet→CUB.** This is a cross-domain scenario, where we employ the training split of mini-ImageNet and the validation and test split of CUB.

**Comparison of Baselines**

As for the description of baseline methods, we refer to Snell & Zemel [27] for a detailed overview. Here we only compare the methods that are most similar to ours, which include DKT, LS (Gibbs), and OVE.

1. **Deep Kernel Transfer (DKT)** [20] utilizes label regression to tackle the conjugacy issue that appeared in classification. In DKT, the multi-class classification problem is transformed into separate binary classification tasks via the one-vs-rest scheme, where labels $\{+1, -1\}$ are treated as continuous values.

2. **Logistic-softmax with Gibbs sampling (LS (Gibbs))** [9] applies the logistic-softmax for a conditional conjugate model after data augmentation. We consider the Gibbs sampling version implemented by Snell & Zemel [27] for Bayesian meta-learning, whose inference method is different from ours. Note that LS (Gibbs) does not use a temperature parameter, which is essentially the scenario of $\tau = 1$ in our notation system.

3. **One-vs-Each Approximation (OVE)** [27] approximates the lower bound of the softmax function for conditional conjugacy after data augmentation. Although it is shown that OVE is a pairwise composite likelihood version of the softmax likelihood, the general approximation capability is weak as shown in the following example. As for implementation, Snell & Zemel [27] utilizes Gibbs sampling as well. We also note that OVE is a transitional invariant likelihood as opposed to the logistic-softmax likelihood.

We present a short example to illustrate the approximation ability of each method in Fig. 1. Here we randomly generate 5,000 samples from $\mathcal{N}(-5, 1)$ for each class and plot the confidence histogram and kernel density estimate of the softmax, Gaussian, logistic-softmax, and OVE likelihood. Apparently, though OVE is an approximation to the lower bound of softmax, it is not similar to softmax classification-wise.

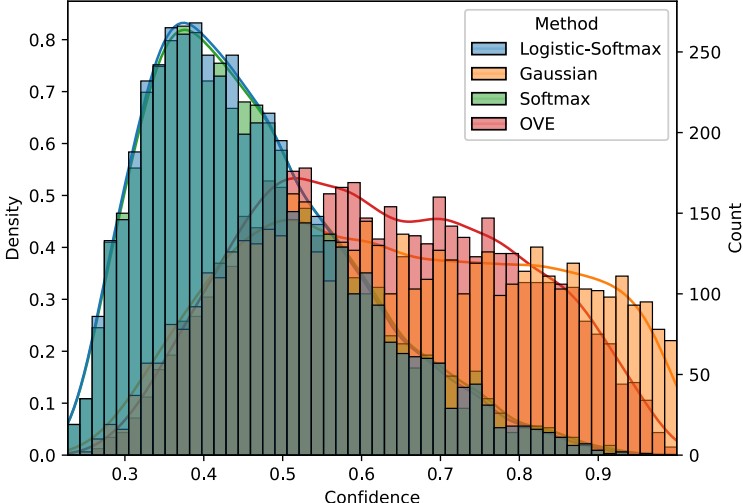

Figure 1: Confidence ($\max_c p(y = c \mid \mathbf{f})$) histogram and kernel density estimate for randomly generated function samples $f_c \sim \mathcal{N}(-5, 1)$. Output probabilities are normalized for $C = 5$.

**Training Protocols**

All of our experiments use the Adam optimizer with a learning rate of $10^{-3}$ for the neural network and a learning rate of $10^{-4}$ for other kernel parameters, following the setting in Patacchiola et al. [20]. During training, all methods use 100 randomly sampled episodes for an epoch. Each episode contains 5 classes and 16 query examples. At test time, 15 query points are evaluated for each episode. We use the validation set to tune all hyperparameters, and the validation set is not applied for training. As for the steps used for mean-field approximation, we run 2 steps during training time and 20 steps during testing time.

**Additional Results**

Now we provide some additional results on different kernels and the training steps of mean-field approximation updates.

In Table 3 we present a comparison between different kernels (Cosine, Linear, Matérn, Polynomial ($p = 1$), Polynomial ($p = 2$), and RBF) trained on 1-shot, ML, ($\tau < 1$) scenarios of CUB and domain transfer. We find that different kernels yield similar results, but the Cosine kernel generally gives a marginally better accuracy across all tasks. This result is in line with both Patacchiola et al. [20] and Snell & Zemel [27].

Table 3: Average 1-shot accuracy and standard deviation on 5-way few-shot classification for different kernels. We use the exact same experiment settings as Cosine for other kernels. Results are evaluated over 5 batches of 600 episodes with different random seeds.

| Method | CUB | CUB→mini-ImageNet |
|---|---|---|
| **Cosine** | $\mathbf{65.21 \pm 0.45}$ | $\mathbf{40.43 \pm 0.43}$ |
| **Linear** | $\mathbf{65.21 \pm 0.50}$ | $39.86 \pm 0.24$ |
| **Matérn** | $64.42 \pm 0.30$ | $39.95 \pm 0.15$ |
| **Polynomial** ($p = 1$) | $64.23 \pm 0.47$ | $39.64 \pm 0.23$ |
| **Polynomial** ($p = 2$) | $64.40 \pm 0.26$ | $39.50 \pm 0.22$ |
| **RBF** | $65.14 \pm 0.50$ | $39.69 \pm 0.18$ |

Additionally, since the task-level update steps of mean-field approximation is a hyperparameter, we investigate the specific effects of different steps. In Fig. 2 we demonstrate a comparison between different steps (starting from 1 to 7) trained on 1-shot, ML, ($\tau < 1$) scenarios of CUB and domain transfer. We find that using 2 or 3 steps is generally optimal, as fewer steps may not lead to convergence and more steps may block the gradient flow of ELBO. As we have mentioned, when the task-level variables are detached from the computational graph, ELBO is not capable of generating an accurate gradient flow for the deep kernel which leads to a collapse in performance.

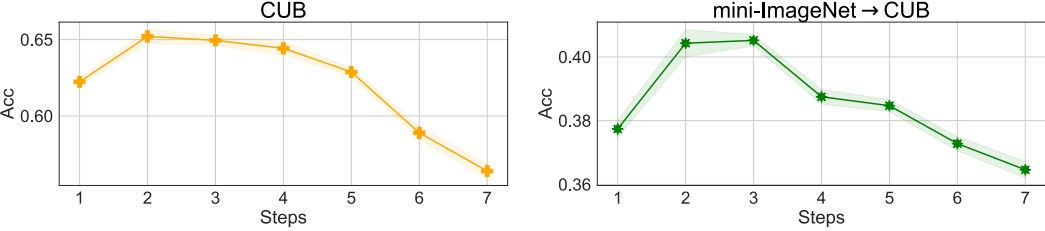

Figure 2: Lineplots of average 1-shot accuracy and standard deviation on 5-way few-shot classification for different steps. We use the exact same experiment settings for all steps. Results are evaluated over 5 batches of 600 episodes with different random seeds.

We also present some additional results with regard to the temperature hyperparameter $\tau$ in logistic-softmax in Table 4. It might be helpful to see the variation in accuracy as the temperature changes.

Table 4: Average 1-shot and 5-shot accuracy and standard deviation on 5-way few-shot classification on CUB. Results are evaluated over 5 batches of 600 episodes with different random seeds.

| Temperature | 0.2 | 0.5 | 0.75 | 1 | 1.5 |
|---|---|---|---|---|---|
| **1-shot** | $\mathbf{65.76 \pm 0.40}$ | $65.16 \pm 0.28$ | $64.02 \pm 0.29$ | $60.85 \pm 0.38$ | $59.43 \pm 0.25$ |
| **5-shot** | $\mathbf{79.10 \pm 0.33}$ | $78.48 \pm 0.18$ | $77.20 \pm 0.13$ | $75.98 \pm 0.33$ | $72.13 \pm 0.20$ |