# OpenReview forum: "Revisiting Logistic-softmax Likelihood in Bayesian Meta-Learning for Few-Shot Classification"
_NeurIPS.cc/2023/Conference — NeurIPS 2023 poster_

### Official Review · Reviewer_1tNK · 2023-06-12

**Soundness:** 3 good
**Presentation:** 3 good
**Contribution:** 3 good
**Rating:** 6
**Confidence:** 3

**Summary:**

This paper tackles the problem that classification tasks in Bayesian meta-learning meet mathematical problems with the softmax function. The previously proposed logistic-softmax function as an alternative can be optimized, but tends to exhibit inherent lack of confidence in prediction. To solve this problem, the authors propose to add a temperature scaling parameter in the function. This simple change not only solves the confidence problem when setting the temperature to less than 1, but also enjoys both theoretical and empirical advantages, as verified in the paper. For efficient and intractable optimization, the authors additionally use the data augmentation technique to derive a fully analytical mean-field inference method for the Bayesian meta-learning model.

**Strengths:**

+ While being simple, the revised logistic-softmax function indeed enjoys better theoretical and empirical advantages, and these are all verified in the paper with proofs or experiments.
+ The proposed logistic-softmax function with temperature has the potential to be applied to broader research areas such as multi-class classification.
+ The writing is clear, making the paper easy to follow.

**Weaknesses:**

- When we just focus on few-shot classification, I have to say that all bayesian meta-learning algorithms are not comparable to even very simple algorithms like ProtoNet or CE Baseline (I mean, if well-tuned, not the performance in Table 1) with the same architecture Conv-4. Also, as standard algorithms now use ResNet-12 as the backbone, all bayesian meta-learning algorithms still use Convnet, making the performance not comparable to those in most recent reported papers. This leads to doubt that do these bayesian meta-learning algorithms still perform well if we use ResNet-12 as the backbone? If the reason not to use ResNet-12 is the computation burden, then the biggest problem may be how to make bayesian inference more tractable. Also, another problem of interest is to figure out why bayesian meta-learning with complicated training steps cannot perform well on few-shot learning, even if equipped with nice theoretical properties. Since this paper does not answer these questions, I think the value of this paper is restricted.


**Questions:**

- In theorem 3.3, $a$ is said to be the mean function with no restriction, so the mean of different sample points can be different. However, in line 444 in the appendix, all different samples have the same mean, can the author explain this?
- How does your method compare with non-Gaussian processes like neural process [1]? It seems that neural process is more flexible and efficient than Gaussian process.

[1] Garnelo et al. Neural processes.

**Limitations:**

The authors have discussed the limitations which I found reasonable. I suggest the authors think more about the role of Gaussian process in bayesian meta-learning, as well as the role of bayesian meta-learning in few-shot classification.

---

> ### Author Rebuttal · Authors · 2023-08-07
>
> Thank you for the valuable advice. As an overview, our paper mainly focuses on two themes: 1. theoretical analysis of
> the logistic-softmax function. 2. application of the logistic-softmax function to Bayesian meta-learning. We admit the second section mainly follows the research line of the Bayesian meta-learning community while paying less attention to the larger context of few-shot classification. Although we do not offer a comprehensive remedy for the problems faced by the Bayesian method in FSC, we believe our theoretical analysis on logistic-softmax and the derivation of the mean-field approximation provides valuable insight and tools for the strive of the Bayesian methods community. In the revised version, we will add more discussions with regard to the role of Gaussian process in Bayesian meta-learning, as well as the role of Bayesian meta-learning in FSC in the Limitations section. Now we answer your questions below.
>
> > Q1. When we just focus on few-shot classification, I have to say that all bayesian meta-learning algorithms are not comparable to even very simple algorithms like ProtoNet or CE Baseline (I mean, if well-tuned, not the performance in Table 1) with the same architecture Conv-4.
>
> Thank you for the comment. In Table 1, the result for ProtoNet is state-of-the-art performance under the same training protocol in the existing literature. Therefore, we may not fully understand the meaning of 'well-tuned' in your concern and would love to engage in further discussion.
>
> In addition, though some Bayesian methods may not be competitive in terms of accuracy, we would like to point out that the major advantage of Bayesian methods is the capability of uncertainty calibration in a probabilistic framework. This is often desirable in fields relying on risk measurement, especially in the context of few-shot classification.
>
>
>
> > Q2. Also, as standard algorithms now use ResNet-12 as the backbone, all Bayesian meta-learning algorithms still use Convnet, making the performance not comparable to those in most recent reported papers. This leads to doubt that do these bayesian meta-learning algorithms still perform well if we use ResNet-12 as the backbone?
>
> Thank you for the question. We would like to emphasize that our work mainly focuses on the theoretical property of modified logistic softmax. Its application to Bayesian meta-learning is largely an attempt to verify our theoretical analysis with empirical results. Therefore, we use the most common backbone structure and training protocol inherited from Bayesian meta-learning. Since OVE and DKT (and other Bayesian meta-learning methods) differ from our method only in their likelihood functions in essence, we use the same Convnet setup as them for better comparison of the logistic-softmax function and others.
>
> Besides, we refer to the paper of DKT [1] which uses both ResNet-10 and Convnet as the backbone and both reaches competitive results. As a similar Bayesian meta-learning method, we believe our method can extend to different neural network structures as well.
>
> [1] Patacchiola, Massimiliano, et al. "Bayesian meta-learning for the few-shot setting via deep kernels." Advances in Neural Information Processing Systems 33 (2020)
>
>
>
> > Q3. If the reason not to use ResNet-12 is the computation burden, then the biggest problem may be how to make Bayesian inference more tractable.
>
> Thank you for raising this question. We agree that the biggest problem is to make Bayesian inference more tractable. In the context of Bayesian meta-learning, the main challenge is to compute the integral for posterior inference. Our derivation of mean-field approximation targets to resolve this problem, as we provide a closed-form expression for variational parameters. In practice, our inference method is more efficient than the Gibbs sampling one while reaching close classification accuracy.
>
>
>
> > Q4. Also, another problem of interest is to figure out why bayesian meta-learning with complicated training steps cannot perform well on few-shot learning, even if equipped with nice theoretical properties. Since this paper does not answer these questions, I think the value of this paper is restricted.
>
> We acknowledge that the theoretical explanation for sub-optimal results for Bayesian methods in meta-learning is not explored in our paper. However, we would like to emphasize that the main focus of this paper is the modified logistic-softmax likelihood, whose good theoretical property invites potential applicability in various domains beyond few-shot learning. Nonetheless, we agree that this is a thrilling and urgent topic and we would love to do some future work for exploration.
>
>
>
> > Q5. In theorem 3.3, $a$ is said to be the mean function with no restriction, so the mean of different sample points can be different. However, in line 444 in the appendix, all different samples have the same mean, can the author explain this?
>
> We really appreciate the reviewer for pointing out this typo and thank the reviewer for the chance for us to clarify it here. The mean function does not have to be constant w.r.t. different sample points. We have proofread the proof of Thm3.3, and our analysis is still correct when the mean function is not constant.
>
>
>
> > Q6. How does your method compare with non-Gaussian processes like neural process [1]? It seems that neural process is more flexible and efficient than Gaussian process.
>
> Thank you for reminding us. The neural process is a promising and efficient method, and we will add some discussion on the extension to non-Gaussian processes such as the neural process in the revised paper.

---

> > ### Comment · Reviewer_1tNK · 2023-08-15
> >
> > Thanks for the authors' response. I'd like to remind the authors that even in the original paper of ProtoNet, the reported accuracy is much better than that in Table 1 of this paper. I maintain the score.

---

> > > ### Author Response · Authors · 2023-08-19
> > >
> > > Thank you for responding to us.
> > >
> > > In terms of your concern, we first notice that in the original paper of ProtoNet, the accuracy result on CUB is in the setting of 50-way and zero-shot, which is different from ours. Therefore, the reported accuracy in the original paper of ProtoNet is not discussed in our paper.
> > >
> > > Moreover, we also notice that the ProtoNet and OVE[1] have the same first author, and OVE recognizes the result we report of ProtoNet on 1-shot 5-way and 5-shot 5-way CUB problems.
> > >
> > > Finally, the result we report of ProtoNet on CUB with 1-shot and 5-shot is first provided by the paper DKT[2] and has been adopted by much literature[1] [3] [4] ever since. All results, as DKT puts in the paper, "are trained from scratch with the same backbone and learning schedule". Therefore, we believe the ProtoNet result in Table 1 is rigorous in our setting. However, we do feel the need to revise our paper to point out this research line in the experiment part. Thank you for reminding us!
> > >
> > >
> > > [1] Snell, Jake, and Richard Zemel. "Bayesian Few-Shot Classification with One-vs-Each Pólya-Gamma Augmented Gaussian Processes." International Conference on Learning Representations. 2020.
> > >
> > > [2] Patacchiola, Massimiliano, et al. "Bayesian meta-learning for the few-shot setting via deep kernels." Advances in Neural Information Processing Systems 33 (2020): 16108-16118.
> > >
> > > [3] Wang, Ze, et al. "Learning to learn dense gaussian processes for few-shot learning." Advances in Neural Information Processing Systems 34 (2021): 13230-13241.
> > >
> > > [4] Sendera, Marcin, et al. "Hypershot: Few-shot learning by kernel hypernetworks." Proceedings of the IEEE/CVF Winter Conference on Applications of Computer Vision. 2023.

---

### Official Review · Reviewer_JdA9 · 2023-07-06

**Soundness:** 3 good
**Presentation:** 3 good
**Contribution:** 3 good
**Rating:** 6
**Confidence:** 3

**Summary:**

The paper revisits the design of logistic-softmax function in the context of classification in Bayesian machine learning. In particular, the paper shows that a logistic-softmax function with a temporature could be more expressive than the conventional softmax function. However, due to the intrinsic nature of the logistic-softmax function, the inference might be non-conjugate, resulting in an intractable solution. To mitigate such an issue, the paper adopts the data augmentation technique proposed in the original logistic-softmax paper and propose a mean-field variational inference as an approximation to make the inference more efficient. The paper demonstrates the capability of the logistic-softmax with temporature in the context of few-shot meta-learning and shows that the approaches that integrate such method could achieve comparable performance in the two classificatio benchmarks: CUB-200-2011 and mini-ImageNet.

**Strengths:**

- The paper has carried out a thorough investigation of the properties of the logisitic-softmax function with a temporature parameter. In particular, the paper shows that the logistic-softmax function with a temporature can easily model a one-hot vector or a uniform one (Theorem 3.1) or it can converge to a softmax function under certain conditions (Theorem 3.2). In addition, the logistic-softmax function of interest could model a richer family of distribution functions (Theorem 3.3).
- The explanation of the paper is clear with easy-to-follow intuitition. The formulations are also clear to increase its clarity.

**Weaknesses:**

The weakness of the paper might be at the applications of the logistic-softmax function. Currently, the paper targets to meta-learning, but to me, the paper is about the logistic-softmax function and meta-learning is just a mere application to demonstrate. Since the authors specify explicitly meta-learning, I cannot ask for other exploration. However, the paper might be strengthen more if it could include other applications using such a function for classification with GP.

**Questions:**

Could the authors discuss further some potential applications using the logistic-softmax function of interest (beside meta-learning mentioned in the paper)?

**Limitations:**

- This is an increamental improvement from the original paper of the logistic-softmax function.
- As mentioned in section 7, the current analysis is carried out in the context of Bayesian meta-learning. It might worth to investigate in other settings.

---

> ### Author Rebuttal · Authors · 2023-08-07
>
> Thank you for your advice on this paper. We answer your questions below:
>
> > Q1. Currently, the paper targets to meta-learning, but to me, the paper is about the logistic-softmax function and meta-learning is just a mere application to demonstrate. However, the paper might be strengthen more if it could include other applications using such a function for classification with GP.
>
> We agree that logistic-softmax is the main focus of this paper. Despite meta-learning being one of its typical applications, we acknowledge that logistic-softmax should be employed in other domains. In future research, we are committed to exploring additional applications of logistic-softmax within and beyond GP classification tasks.
>
>
>
> > Q2. Could the authors discuss further some potential applications using the logistic-softmax function of interest (besides meta-learning mentioned in the paper)?
>
> Thank you for raising this question. We believe our logistic-softmax function has the potential to replace the softmax function in various domains. To begin with, our function can be applied to Gaussian process classification tasks, including class-imbalanced scenarios [1], active learning [2], and time-series data analysis [3]. In this case, logistic-softmax brings desired conditional conjugacy to make inference tractable and provides additional flexibility in data modeling than softmax as indicated in our paper. Moreover, the logistic-softmax function can be a great choice in modern Bayesian methods, such as Bayesian neural networks and neural network Gaussian processes although further adaptation is needed. Furthermore, our logistic-softmax function might be capable of replacing softmax beyond the Bayesian domain since we prove its flexibility over the softmax function. For example, as the logistic-softmax function captures positive signals for multiple classes, it may have prospective advantages in scenarios like multi-label classification [4] and multi-label contrastive learning [5], ushering in new paradigms thanks to its unique property.
>
> [1] Ye, Changkun, et al. "Efficient Gaussian Process Model on Class-Imbalanced Datasets for Generalized Zero-Shot Learning." 2022 26th International Conference on Pattern Recognition (ICPR). IEEE, 2022.
>
> [2] Zhao, Guang, et al. "Efficient active learning for Gaussian process classification by error reduction." Advances in Neural Information Processing Systems 34 (2021)
>
> [3] Constantin, Alexandre, Mathieu Fauvel, and Stéphane Girard. "Mixture of multivariate gaussian processes for classification of irregularly sampled satellite image time-series." Statistics and Computing 32.5 (2022)
>
> [4] Lanchantin, Jack, et al. "General multi-label image classification with transformers." Proceedings of the IEEE/CVF Conference on Computer Vision and Pattern Recognition. 2021.
>
> [5] Zhang, Shu, et al. "Use all the labels: A hierarchical multi-label contrastive learning framework." Proceedings of the IEEE/CVF Conference on Computer Vision and Pattern Recognition. 2022.
>
>
>
> > Q3. This is an incremental improvement from the original paper of the logistic-softmax function.
>
> We acknowledge that our work is built upon the original paper on logistic-softmax, but we would like to emphasize that the main theoretical findings of our research have not been explored in the existing literature to the best of our knowledge. Furthermore, the application of logistic-softmax in Bayesian meta-learning with mean-field posterior inference is a novel contribution of our work. Through empirical evaluation, we have achieved state-of-the-art performance on several benchmarks, providing empirical validation for our theoretical analysis. While we admit the modification is simple, we believe our theoretical and methodological findings support the significance and relevance of our research.

---

> > ### Comment · Reviewer_JdA9 · 2023-08-14
> > **Discussion**
> >
> > Thank you, the authors, for addressing my concerns.

---

> > > ### Author Response · Authors · 2023-08-14
> > > **Thank you for your quick response**
> > >
> > > Thank you for your quick response. With all your concerns now resolved, would you be willing to consider increasing the score?

---

> > > > ### Comment · Reviewer_JdA9 · 2023-08-20
> > > > **Assessment after rebuttal**
> > > >
> > > > I maintain my assessment after reading the authors' rebuttal.

---

### Official Review · Reviewer_L2f9 · 2023-07-06

**Soundness:** 4 excellent
**Presentation:** 3 good
**Contribution:** 3 good
**Rating:** 6
**Confidence:** 3

**Summary:**

This paper proposes to modify the logistic Softmax likelihood by including a temperature coefficient in GP-based meta learning for few-shot classification. This is motivated by the observation that prediction made by logistic Softmax likelihood do not have confidence.
Theoretically, they demonstrate that using the temperature, one could control for the confidence in logistic softmax. Furthermore, it is proved that softmax likelihood is a special case of logistic softmax. They also propose to use mean-field variational inference to approximate the posterior which is computationally more efficient than the typically adapted Gibbs sampler. It is demonstrated that their method achieves superior performance in uncertainty quantification. However, the performance improvements measured by accuracy are not as impressive as the uncertainty quantification.


**Strengths:**

The problem is well motivated, and all the required theoretical derivations have been included.
There are new theoretical findings presented.

**Weaknesses:**

The idea of including temperature is not novel but its application to a new domain is novel.
Including one more recent method for few-shot classification, referenced in my questions, can improve the benchmark.

**Questions:**

Q1. I understand that the typical benchmark for few-shot classification uses 5 classes. Have you seen improvements with your method when there are more than 5 classes? Like 10 or 20? (Answering this question won’t impact my decision negatively.)

Q2. Can you also include https://arxiv.org/pdf/2101.06395.pdf method in your benchmark?

Q3. I understand that the temperature has been tuned for your experiments, but it would be nice to see the variation in the accuracy and uncertainty estimation as the temperature changes. Can you demonstrate this relationship for a subset of your experiments?

---

> ### Author Rebuttal · Authors · 2023-08-07
>
> Thank you so much for your advice to this paper. We answer your questions below:
>
> > Q1. I understand that the typical benchmark for few-shot classification uses 5 classes. Have you seen improvements with your method when there are more than 5 classes? Like 10 or 20? (Answering this question won’t impact my decision negatively.)
>
> We understand your concern about our method's effectiveness on 10 or 20-way few-shot problems. We acknowledge that we have not run experiments on those settings because most benchmarks in Bayesian meta-learning do not include results for 10 or 20-way few-shot problems. In future research, we will investigate the potential impact brought by the choice of class numbers, but we believe more classes will not fundamentally change the effect of our current method.
>
>
>
> > Q2. Can you also include https://arxiv.org/pdf/2101.06395.pdf method in your benchmark?
>
> The referred paper proposes a novel idea for few-shot learning by calibrating data distribution. Although the method brings promising results on similar benchmarks, we suppose its framework is a bit different from the Bayesian meta-learning methods considered in this paper. In fact, the calibration method is compatible with our proposed method since we essentially introduce a Bayesian framework to train classifiers, and the calibration method works with arbitrary classifiers. However, we would love to add some discussion about the referred paper to give a more comprehensive view of typical few-shot classification methods.
>
>
>
> > Q3. I understand that the temperature has been tuned for your experiments, but it would be nice to see the variation in the accuracy and uncertainty estimation as the temperature changes. Can you demonstrate this relationship for a subset of your experiments?
>
> Thank you for the question. We have added some results on the CUB dataset with different temperature parameters.
> We hope the additional empirical evidence resolves your concern.
>
> | Temperature | 0.2              | 0.5              | 0.75             | 1                | 1.5              |
> | ----------- | ---------------- | ---------------- | ---------------- | ---------------- | ---------------- |
> | CUB 1 shot  | 65.76 $\pm$ 0.40 | 65.16 $\pm$ 0.28 | 64.02 $\pm$ 0.29 | 60.85 $\pm$ 0.38 | 59.43 $\pm$ 0.25 |
> | CUB 5 shot  | 79.10 $\pm$ 0.33 | 78.48 $\pm$ 0.18 | 77.20 $\pm$ 0.13 | 75.98 $\pm$ 0.33 | 72.13 $\pm$ 0.20 |

---

> > ### Comment · Reviewer_L2f9 · 2023-08-12
> >
> > Thanks for your response. All my questions have been addressed. I will maintain my original score.

---

> > > ### Author Response · Authors · 2023-08-12
> > > **Thank you for your quick response**
> > >
> > > Thank you for your quick response. With all your concerns now resolved, would you be willing to consider increasing the score?

---

### Official Review · Reviewer_iEEK · 2023-07-29

**Soundness:** 3 good
**Presentation:** 3 good
**Contribution:** 3 good
**Rating:** 5
**Confidence:** 3

**Summary:**

In this work, the logistic-softmax likelihood is redesigned, allowing control of the a priori confidence level through a temperature parameter. The modified logistic-softmax is shown to encompass softmax as a special case and induces a larger family of data distributions. By integrating this modified likelihood into a deep kernel-based Gaussian process meta-learning framework with data augmentation, well-calibrated uncertainty estimates are achieved in experiments, and competitive results are obtained on standard benchmark datasets.

Post rebuttal: I have read the authors' rebuttal and I appreciate the authors' effort in addressing my concerns.

**Strengths:**

+The logistic-softmax function with temperature is a nice idea and has potential to be used in multiple domains.
+The theoretical analysis of the logistic-softmax likelihood is solid.
+Some promising results are presented.
+The paper is well written and easy to follow.

**Weaknesses:**

-The proposed modified logistic-softmax function with temperature can be fundamental for different machine learning problems, it is not clear why it is specifically used for Bayesian meta-learning for few-shot classification. It is suggested to be tested on simpler tasks first before applying it to tasks like Bayesian meta-learning.
-The proposed method performs marginally worse than state-of-the-art in maximum calibration error on a few benchmark datasets. It is not clear why how it happens. Theoretically, an adaptive temperature is expected to give better performance for all the tasks.

**Questions:**

Q1: Have you considered other forms of the temperature in the modified logistic-softmax likelihood, for example having it as the power of the functions?
Q2: Can you give a bit more detail of how the modified logistic-softmax function in multi-label classification?

**Limitations:**

The limitations have been discussed in the paper: the performance of the proposed modified logistic-softmax is only evaluated in Bayesian meta-learning.

---

> ### Author Rebuttal · Authors · 2023-08-07
>
> Thank you so much for reviewing our paper. We answer your questions below.
>
> > Q1. The proposed modified logistic-softmax function with temperature can be fundamental for different machine learning problems, it is not clear why it is specifically used for Bayesian meta-learning for few-shot classification.It is suggested to be tested on simpler tasks first before applying it to tasks like Bayesian meta-learning.
>
> Thank you so much for pointing this out. Logistic-softmax was initially proposed to address the conjugation issue that arises in multi-class Gaussian process classification. As the Bayesian framework is advantageous in uncertainty calibration, some researchers focus on adapting the multi-class Gaussian process to few-shot classification tasks, where Bayesian meta-learning is one of the prevalent paradigms. In specific, OVE attempts to apply it to Bayesian meta-learning but fails to achieve optimal results. Therefore, we specify Bayesian meta-learning because it is one of the initial applications for the original logistic-softmax likelihood in literature and is consistent with our research line. Nonetheless, our theoretical analysis has broad applicability across various domains beyond its motivation from the Bayesian framework.
>
> Meanwhile, we do realize that the presentation of our paper could be adjusted for better clarity and we thank you for pointing this out. We will elaborate on why we apply the logistic-softmax function to Bayesian meta-learning and explain our motivation more clearly in the revised paper. In addition, we will add more discussions on other potential applications of logistic-softmax.
>
>
>
> > Q2. The proposed method performs marginally worse than state-of-the-art in maximum calibration error on a few benchmark datasets. It is not clear why how it happens. Theoretically, an adaptive temperature is expected to give better performance for all the tasks.
>
> Thank you for raising this concern. We notice that our result on maximum calibration error is worse in only one dataset. One possible explanation for this phenomenon is that the adaptive temperature is tuned on the validation set (line 295). It is highly possible that in the test dataset, there are some outliers that this particular temperature is unable to effectively handle. To provide some context, the MCE result of 0.036 is from a bin with a small number of samples.
>
>
>
> > Q3. Have you considered other forms of the temperature in the modified logistic-softmax likelihood, for example having it as the power of the functions?
>
> Thank you for reminding us, we will add a section discussing possible forms of temperature in the revised paper. We consider the current form because we are motivated by temperature in contrastive learning, where the most popular choice is to scale the logit directly. In addition, we also believe that it is the simplest way to maintain the conditional conjugate structure, while using the power of the function may lead to an intractable integral unless further adapted.
>
>
>
> > Q4. Can you give a bit more detail of how the modified logistic-softmax function in multi-label classification?
>
> Thank you for raising this question. In general, multi-class classification is often treated as a combination of binary classification problems. For example, [1-3] use independent classifiers for each label, and design a sum of binary cross-entropy(BCE) loss as the training objective. From a higher level, this phenomenon is due to the lack of an appropriate likelihood function that identifies multiple positive labels at the same time, and researchers have to rely on the One-vs-Rest scheme. Specifically, if we use the softmax function (at a low temperature) to process the logits, the output fails to depict the probability of multiple labels, as softmax converges to a one-hot vector (for example, the positive label includes cat and dog, logits are 5 and 4.9, softmax (at a low temperature) outputs 0.99 and 0.01). With our modified logistic-softmax function, it can leverage the upside of a low temperature while preserving an appropriate probability output (logits are 5 and 4.9, logistic-softmax (at a low temperature) outputs 0.51 and 0.49). Although we acknowledge the potential barriers in the application to multi-class classification (e.g., designing novel loss function paired with logistic-softmax), we hold a strong belief in its potential to motivate new paradigms in multi-label classification.
>
> [1] Lanchantin, Jack, et al. "General multi-label image classification with transformers." Proceedings of the IEEE/CVF Conference on Computer Vision and Pattern Recognition. 2021.
>
> [2] Wang, Haoran, et al. "Can multi-label classification networks know what they don’t know?." Advances in Neural Information Processing Systems 34 (2021)
>
> [3] Panos, Aristeidis, Petros Dellaportas, and Michalis K. Titsias. "Large scale multi-label learning using Gaussian processes." Machine Learning 110 (2021)

---

> > ### Comment · Reviewer_iEEK · 2023-08-15
> > **Response acknowledged**
> >
> > I thank the authors for providing the response

---

> > > ### Author Response · Authors · 2023-08-15
> > > **Thank you for your response**
> > >
> > > Thank you for your response. If you have no other concerns, would you be willing to consider increasing the score?

---

### Author Rebuttal · Authors · 2023-08-07

We extend our sincere appreciation to all reviewers for their time, effort, and insightful feedback. We are encouraged by their recognition of the significance of our work in introducing an effective modification to control the confidence of logistic-softmax, uncovering novel theoretical properties and broad applicability, deriving an efficient mean-field inference method, conducting comprehensive numerical experiments, and maintaining clear and concise writing.

In the following, we respond meticulously to each of the reviewers' comments. Our aim is to ensure that we address all the concerns and offer clarity and reassurance where needed. Should any additional questions arise, we invite reviewers to engage in further discussion. Once again, we express our gratitude for your time and dedication in reviewing our work.

---

### Decision · Program_Chairs · 2023-09-21

**Decision:**

Accept (poster)

**Comment:**

The submission revisits the logistic-softmax likelihood in the context of Bayesian meta-learning for few-shot classification. It proposes a modification which incorporates a temperature coefficient to address the lack of confidence inherent to the approach. It also sidesteps intractability issues using a mean-field variational inference approximation to make inference more efficient. Test episode query accuracies are presented for CUB, mini-ImageNet, and mini-ImageNet → CUB for 1-shot and 5-shot classification, along with expected calibration error (ECE) and maximum calibration error (MCE).

Reviewers note that the submission is clear and well-written (iEEK, JdA9) and presents a broadly-applicable idea (iEEK) showing promising results (iEEK) and backed up by a solid theoretical analysis (iEEK, L2f9, JdA9).

The most salient reviewers concerns are:

* Reviewers iEEK and JdA9 would like to see the idea applied in settings beyond Bayesian meta-learning for few-shot classification. The authors explain that they chose this setting as it is "one of the initial applications for the original logistic-softmax likelihood in the literature" and that they will clarify this point in the submission. This addresses Reviewer JdA9's concern.
* Reviewer iEEK notes that the proposed approach does not consistently outperform competing approaches in terms of MCE. The authors acknowledge that this is the case for one dataset, which could be the result of a mismatch between the validation set and outliers in the test set. Their response sounds reasonable to me: I think what's more important here is the overall pattern of strong results.
* Reviewer L2f9 would like to see a comparison against Distribution Calibration, and Reviewer 1tNK points out that Bayesian meta-learning approaches still standardize around the four-layer convnet architecture, whereas other few-shot classification works have moved on to ResNet-12 and other higher-capacity architectures. The authors explain that Bayesian methods are not necessarily competitive in terms of accuracy, but they have uncertainty calibration capabilities which other few-shot classification approaches do not. To me, this is a reasonable justification for the scope of competing approaches to compare against.
* Reviewer 1tNK is also concerned that the original Prototypical Networks paper presents stronger numbers than what is reported in the submission. The authors clarify that they report results that are presented in the OVE paper, whose first author is also the first author of the Prototypical Networks paper.

Overall all reviewers are in favor of acceptance (albeit weakly), and the authors' response addresses the reviewers' concerns adequately from my vantage point.